# POPULATION-BASED REINFORCEMENT LEARNING FOR COMBINATORIAL OPTIMIZATION PROBLEMS

## ABSTRACT

Applying reinforcement learning (RL) to combinatorial optimization problems is attractive as it removes the need for expert knowledge or pre-solved instances. However, it is unrealistic to expect an agent to solve these (often NP-)hard problems in a single shot at inference due to their inherent complexity. Thus, leading approaches often implement additional search strategies, from stochastic sampling and beam-search to explicit fine-tuning. In this paper, we argue for the benefits of learning a population of complementary policies, which can be simultaneously rolled out at inference. To this end, we introduce Poppy, a simple training procedure for populations. Instead of relying on a predefined or hand-crafted notion of diversity, Poppy induces an unsupervised specialization targeted solely at maximizing the performance of the population. We show that Poppy produces a set of complementary policies, and obtains state-of-the-art RL results on three popular NP-hard problems: the traveling salesman (TSP), the capacitated vehicle routing (CVRP), and 0-1 knapsack (KP) problems. On TSP specifically, Poppy outperforms the previous state-of-the-art, dividing the optimality gap by 5 while reducing the inference time by more than an order of magnitude.

## 1 INTRODUCTION

In recent years, machine learning (ML) approaches have overtaken algorithms that use handcrafted features and strategies across a variety of challenging tasks (Mnih et al., 2015; van den Oord et al., 2016; Silver et al., 2017; Brown et al., 2020). In particular, solving combinatorial optimization (CO) problems – where the maxima or minima of an objective function acting on a finite set of discrete variables is sought – has attracted significant interest (Bengio et al., 2021) due to both their (often NP) hard nature and numerous practical applications across domains varying from logistics (Sbihi & Eglese, 2007) to fundamental science (Wagner, 2020).

As the search space of feasible solutions typically grows exponentially with the problem size, exact solvers can be challenging to scale, hence CO problems are often also tackled with handcrafted heuristics using expert knowledge. Whilst a diversity of ML-based heuristics have been proposed, reinforcement learning (RL; Sutton & Barto, 2018) is a promising paradigm as it does not require pre-solved examples of these hard problems. Indeed, algorithmic improvements to RL-based CO solvers, coupled with low inference cost, and the fact that they are by design targeted at specific problem distributions, have progressively narrowed the gap with traditional solvers.

To improve the quality of proposed solutions, RL methods typically generate multiple candidates with additional search procedures, which can be divided into two families (Mazyavkina et al., 2021). First, *improvement methods* start from a feasible solution and iteratively improve it through small modifications (actions). However, such incremental search cannot quickly access very different solutions, and requires handcrafted procedures to define a sensible action space. Second, *construction methods* incrementally build a solution by selecting one element at a time. Multiple solutions can be built using sampling strategies, such as stochastic sampling policies or beam search. However, just as improvement methods are biased by the initial starting solution, construction methods are biased by the single underlying policy. Thus, a balance must be struck between the exploitation of the learned policy (which may be ill-suited for a given problem instance) and the exploration of different solutions (where the extreme case of a purely random policy will likely be highly inefficient).

In this work, we propose Poppy, a construction method that uses a *population* of agents with suitably diverse policies to improve the exploration of the solution space of hard CO problems. Whereas a single agent aims to perform well across the entire problem distribution, and thus has to make compromises, a population can learn a set of heuristics such that only one of these has to be performant on any given problem instance. However, realizing this intuition presents several challenges: (i) naïvely training a population of agents is expensive and challenging to scale, (ii) the trained population should have complementary policies that propose different solutions, and (iii) the training approach should not impose any handcrafted notion of diversity within the set of policies given the absence of clear behavioral markers aligned with performance for typical CO problems.

Challenge (i) is addressed by sharing a large fraction of the computations across the population, specializing only lightweight policy heads to realize the diversity of agents. Challenges (ii) and (iii) are jointly achieved by introducing an RL objective aimed at specializing agents on distinct subsets of the problem distribution. Concretely, we derive a lower bound of the true population-level objective, which corresponds to training only the agent which performs best on each problem. This is intuitively justified as the performance of the population on a given problem is not improved by training an agent on an instance where another agent already has better performance. Strikingly, we find that judicious application of this conceptually simple objective gives rise to a population where the diversity of policies is obtained without explicit supervision (and hence is applicable across a range of problems without modification) and essential for strong performance.

Our contributions are summarized as follows:

1. We motivate the use of populations for CO problems as an efficient way to explore environments that are not reliably solved by single-shot inference.

2. We derive a new training objective and present a practical training procedure that encourages performance-driven diversity (i.e. effective diversity without the use of explicit behavioral markers or other external supervision).

3. We evaluate Poppy on three CO problems: TSP, CVRP, and 0-1 knapsack (KP). On TSP and KP, Poppy significantly outperforms other RL-based approaches. On CVRP, it consistently outperforms other inference-only methods and approaches the performance of actively fine-tuning problem-specific policies.

## 2 RELATED WORK

**ML for Combinatorial Optimization** The first attempt to solve TSP with neural networks is due to Hopfield & Tank (1985), which only scaled up to 30 cities. Recent developments of bespoke neural architectures (Vinyals et al., 2015; Vaswani et al., 2017) and performant hardware have made ML approaches increasingly efficient. Indeed, several architectures have been used to address CO problems, such as graph neural networks (Dai et al., 2017), recurrent neural networks (Nazari et al., 2018), and attention mechanisms (Deudon et al., 2018). In this paper, we use an encoder-decoder architecture that draws from that proposed by Kool et al. (2019). The costly encoder is run once per problem instance, and the resulting embeddings are fed to a small decoder iteratively rolled out to get the whole trajectory, which enables efficient inference. This approach was furthered by Kwon et al. (2020), who leveraged the underlying symmetries of typical CO problems (e.g. of starting positions and rotations) to realize improved training and inference performance using instance augmentations. Kim et al. (2021) also draws on Kool et al. and uses a hierarchical strategy where a seeder proposes solution candidates, which are refined bit by bit by a reviser. Closer to our work, Xin et al. (2021) trains multiple policies using a shared encoder and separate decoders. Whilst this work (MDAM) shares our architecture and goal of training a population, our approach for enforcing diversity differs substantially. MDAM explicitly trades off performance with diversity by jointly optimizing policies and their KL divergence. Moreover, as computing the KL divergence for the whole trajectory is intractable, MDAM is restricted to only using it to drive diversity at the first timestep. In contrast, Poppy drives diversity by maximizing population-level performance (i.e. without any explicit diversity metric), uses the whole trajectory and scales better with the population size (we have used up to 32 agents instead of only 5).

Additionally, ML approaches usually rely on mechanisms to generate multiple candidate solutions (Mazyavkina et al., 2021). One such mechanism consists in using improvement methods on an

initial solution: de O. da Costa et al. (2020) uses policy gradients to learn a policy that selects local operators (2-opt) given a current solution in TSP, while Lu et al. (2020) and Wu et al. (2021) extend this method to CVRP. This idea has been extended to enable searching a learned latent space of solutions (Hottung et al., 2021). However, these approaches have two limitations: they are environment-specific, and the search procedure is inherently biased by the initial solution.

An alternative exploration mechanism is to generate a diverse set of trajectories by stochastically sampling a learned policy, potentially with additional beam search (Joshi et al., 2019), Monte Carlo tree search (Fu et al., 2021), dynamic programming (Kool et al., 2021) or active search (Hottung et al., 2022). However, intuitively, the generated solutions tend to remain close to the underlying deterministic policy, implying that the benefits of additional sampled candidates diminish quickly.

**Population-Based RL**   Populations have already been used in RL to learn diverse behaviors. In a different context, Gupta et al. (2018), Eysenbach et al. (2019), Hartikainen et al. (2020) and Pong et al. (2020) use a single policy conditioned on a set of goals as an implicit population for unsupervised skill discovery. Closer to our approach, another line of work revolves around explicitly storing a set of distinct policy parameters. Doan et al. (2020), Hong et al. (2018), Jung et al. (2020) and Parker-Holder et al. (2020) use a population to achieve a better coverage of the policy space. . However, they enforce explicit attraction-repulsion mechanisms, which is a major difference with respect to our approach where diversity is a pure byproduct of performance optimization.

Our method is also related to approaches combining RL with evolutionary algorithms (EA; Khadka & Tumer, 2018; Khadka et al., 2019; Pourchot & Sigaud, 2019), which benefit from the sample-efficient RL policy updates while enjoying evolutionary population-level exploration. However, the population is a means to learn a unique strong policy, whereas Poppy learns a set of complementary strategies. More closely related, Quality-Diversity (QD; Pugh et al., 2016; Cully & Demiris, 2018) is a popular EA framework which maintains a portfolio of diverse policies. Pierrot et al. (2022) has recently combined RL with a QD algorithm, Map-Elites (Mouret & Clune, 2015); unlike Poppy, QD methods rely on handcrafted behavioral markers, which is not easily amenable to the CO context.

One of the drawbacks of population-based RL is its expensive cost. However, recent approaches have shown that modern hardware as well as targeted frameworks enable efficient vectorized population training (Flajolet et al., 2022), opening the door to a wider range of applications.

## 3 METHODS

### 3.1 BACKGROUND AND MOTIVATION

**RL Formulation**   A CO problem instance $\rho$ sampled from some distribution $\mathcal{D}$ consists of a discrete set of $N$ variables (e.g. city locations in TSP). We model a CO problem as a Markov decision process (MDP) defined by a state space $\mathcal{S}$, an action space $\mathcal{A}$, a transition function $T$, a reward function $R$ and a discount factor $\gamma$. A state is a trajectory through the problem instance $\tau_t = (x_1, \ldots, x_t) \in \mathcal{S}$ where $x_i \in \rho$, and thus consists of an ordered list of variables (not necessarily of length $N$). An action, $a \in \mathcal{A} \subseteq \rho$, consists of choosing the next variable to add; thus, given state $\tau_t = (x_1, \ldots, x_t)$ and action $a$, the next state is $\tau_{t+1} = T(\tau_t, a) = (x_1, \ldots, x_t, a)$. Let $\mathcal{S}^* \subseteq \mathcal{S}$ be the set of *solutions*; that is, states that comply with the problem's constraints (e.g., a sequence of cities such that each city is visited once and ends with the starting city in TSP). The reward function $R : \mathcal{S}^* \to \mathbb{R}$ maps solutions into scalars. We assume the reward is maximized by the optimal solution (e.g. $R$ returns the negative tour length in TSP).

A *policy* $\pi_\theta$ parameterized by $\theta$ can be used to generate solutions for any instance $\rho \sim \mathcal{D}$ by iteratively sampling the next action $a \in \mathcal{A}$ according to the probability distribution $\pi_\theta(\cdot \mid \rho, \tau_t)$. We learn $\pi_\theta$ using REINFORCE (Williams, 1992). This method aims at maximizing the RL objective $J(\theta) \doteq \mathbb{E}_{\rho \sim \mathcal{D}} \mathbb{E}_{\tau \sim \pi_\theta, \rho} R(\tau)$ by adjusting $\theta$ such that good trajectories are more likely to be sampled in the future. Formally, the policy parameters $\theta$ are updated by gradient ascent using $\nabla_\theta J(\theta) = \mathbb{E}_{\rho \sim \mathcal{D}} \mathbb{E}_{\tau \sim \pi_\theta, \rho} (R(\tau) - b_\rho) \nabla_\theta \log(p_\theta(\tau))$, where $p_\theta(\tau) = \prod_t \pi_\theta(a_{t+1} \mid \rho, \tau_t)$ and $b_\rho$ is a baseline. The gradient of the objective, $\nabla_\theta J$, can be estimated empirically using Monte Carlo simulations.

**Motivating Example**   We argue for the benefits of training a population using the example in Figure 1. In this environment, there are three actions: **Left**, **Right**, and **Up**. **Up** leads to a medium

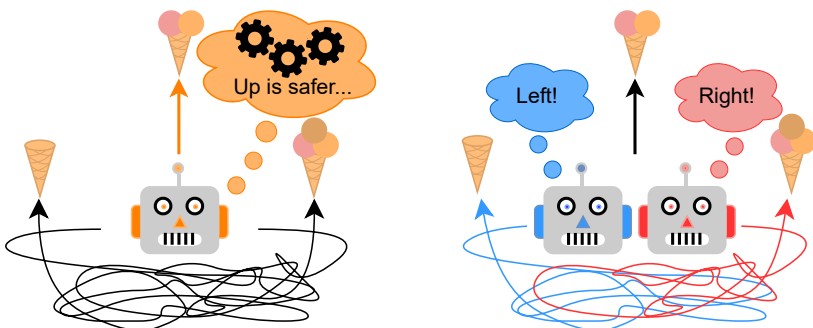

Figure 1: In this environment, the upward path always leads to a medium reward, while the left and right paths are intricate such that either one may lead to a low reward or high reward with equal probability. **Left**: An agent trained to maximize a sum of rewards will converge to taking the safe upward road since it does not have enough information to act optimally. **Right**: A two-agent population can always try the left and right paths and thus get the largest reward in any instance.

reward, while **Left/Right** lead to low/high or high/low rewards (the configuration is determined with equal probability at the start of each episode). Crucially, the left and right paths are intricate, so the agent cannot easily infer from its observation which one leads to a higher reward. Then, the best strategy for a single agent is to always go **Up**, as the guaranteed medium reward (2 scoops) is higher than the expected reward of guessing left or right (1.5 scoops). In contrast, two agents in a population can go in opposite directions and always find the maximum reward. There are two striking observations: (i) the agents do not need to perform optimally for the population performance to be optimal (one agent gets the maximum reward), and (ii) the average performance is worse than in the single-agent case.

Specifically, the discussed phenomenon can occur when (i) some optimal actions are too difficult to infer from observations and (ii) choices are irreversible (i.e. it is not possible to recover from a suboptimal decision). These conditions usually hold when solving hard CO problems. In these situations, as shown above, maximizing the performance of a population will require agents to specialize and likely yield better results than in the single agent case.

### 3.2 POPPY

We present the three distinct components of Poppy: an architecture enabling efficient population rollouts, an RL objective encouraging agent specialization, and the overall training procedure.

**Architecture** We use the attention model introduced by Kool et al. (2019), which decomposes the policy model into two parts. First, a large encoder $h_\psi$, takes an instance $\rho$ as input and outputs embeddings $\omega$ for each of the variables in $\rho$. Second, a smaller decoder $q_\phi$ takes the embeddings $\omega$ and a trajectory $\tau_t$ as input, and outputs the probabilities of each of the next possible actions. Crucially, the expensive computation of embeddings can be done once for every instance at the beginning of an episode since it is independent of any trajectories; hence, only lightweight decoding operations are performed at each timestep.

We exploit this framework to build a population of $K$ agents. The encoder $h_\psi$ is shared as a common backbone for the whole population, whereas the decoders $q_{\phi_1}, q_{\phi_2}, \ldots, q_{\phi_K}$ are unique to each agent. This is motivated by (i) the encoder learning general representation useful for all agents, and (ii) reducing the overhead of training a population and keeping the total number of parameters low. A discussion on the model sizes is provided in Appendix A.1.

**Population-Based Training Objective** The usual RL objective $J(\theta) = \mathbb{E}_{\rho \sim \mathcal{D}} \mathbb{E}_{\tau \sim \pi_\theta, \rho} R(\tau)$ was previously presented in Section 3.1. Intuitively, the *population objective* corresponds to rolling out every agent in the population for each problem and using the best solution. For the parameters $\bar{\theta} = \{\theta_1, \theta_2, \ldots, \theta_K\}$ of a population of $K$ agents, the population objective can be thus defined as:

$$J_{\text{pop}}(\bar{\theta}) \doteq \mathbb{E}_{\rho \sim \mathcal{D}} \mathbb{E}_{\tau_1 \sim \pi_{\theta_1}, \tau_2 \sim \pi_{\theta_2} \ldots, \tau_K \sim \pi_{\theta_K}} \max \left[ R(\tau_1), R(\tau_2), \ldots, R(\tau_K) \right].$$

In order to update $\bar{\theta}$, we derive a simple lower bound designed at inducing agent specialization:

$$J_{\text{pop}}(\bar{\theta}) = \mathbb{E}_{\rho \sim \mathcal{D}} \mathbb{E}_{\tau_1 \sim \pi_{\theta_1}, \tau_2 \sim \pi_{\theta_2}, \ldots, \tau_K \sim \pi_{\theta_K}} \max \left[ R(\tau_1), R(\tau_2), \ldots, R(\tau_K) \right],$$

$$J_{\text{pop}}(\bar{\theta}) \geq \mathbb{E}_{\rho \sim \mathcal{D}} \max \left[ \mathbb{E}_{\tau_1 \sim \pi_{\theta_1}} R(\tau_1), \mathbb{E}_{\tau_2 \sim \pi_{\theta_2}} R(\tau_2), \ldots, \mathbb{E}_{\tau_K \sim \pi_{\theta_K}} R(\tau_K) \right],$$

$$J_{\text{pop}}(\bar{\theta}) \geq \mathbb{E}_{\rho \sim \mathcal{D}} \left[ \mathbb{E}_{\tau \sim \pi_{\theta_\rho^*}} R(\tau) \right] \text{ with } \theta_\rho^* = \arg\max_{\theta_i \in \bar{\theta}} \mathbb{E}_{\tau \sim \pi_{\theta_i}, \rho} R(\tau).$$

This lower bound, which we denote $J_{\text{poppy}}$ and treat as our RL objective, corresponds to evaluating only the best performing agent on each instance. Importantly, $J_{\text{poppy}}$ can be optimized using standard policy gradients (Sutton et al., 1999), with the difference that for each problem only the best agent is trained. As we do not have access to $\theta_\rho^*$, we estimate these as those of the agent with the highest reward on $\rho$ (breaking ties arbitrarily). This leads to an approximated optimization of $J_{\text{poppy}}$. We analyze the resulting bias in the gradient estimation of the objective in Appendix B, and find that it vanishes in the limits of either deterministic agents (where only a single trajectory per agent is required to know the best policy) or perfect specializations (where each agent will always outperform the entire rest of the population on instances to which it is specialized). Intuitively and empirically, we therefore observe that the gradient bias reduces over the course of training. This formulation is applicable across a variety of problems and directly optimizes for population-level performance without explicit supervision or handcrafted behavioral markers.

We emphasize that optimizing the presented objective does not provide any strict diversity guarantee. However, note that diversity maximizes our objective in the highly probable case that, within the bounds of finite capacity and training, a single agent is not optimal on all subsets of the training distribution. Therefore, intuitively (and, as we will show, practically) diversity emerges over training in the pursuit of maximizing the objective.

---

**Algorithm 1:** Poppy training

**input:** problem distribution $\mathcal{D}$, number of agents $K$, batch size $B$, number of training steps $H$, $K$ sets of parameters $q_{\phi_1}, q_{\phi_2}, \ldots, q_{\phi_K}$ ;

**for** *step 1 to H* **do**

    $\rho_i \leftarrow \texttt{Sample}(\mathcal{D}) \, \forall i \in 1, \ldots, B$ ;

    $\tau_i^k \leftarrow \texttt{Rollout}(\rho_i, q_{\phi_k}) \, \forall i \in 1, \ldots, B, \forall k \in 1, \ldots, K$ ;

    `/* Select the best agent for each problem` $\rho_i$`. */`

    $k_i^* \leftarrow \arg\max_{k \leq K} R(\tau_i^k) \, \forall i \in 1, \ldots, B$ ;

    `/* Propagate the gradients through these only. */`

    $\nabla L(q_{\phi_1}, q_{\phi_2}, \ldots, q_{\phi_K}) \leftarrow \frac{1}{B} \sum_{i \leq B} \texttt{REINFORCE}(\tau_i^{k_i^*})$ ;

    $(q_{\phi_1}, q_{\phi_2}, \ldots, q_{\phi_K}) \leftarrow (q_{\phi_1}, q_{\phi_2}, \ldots, q_{\phi_K}) - \alpha \nabla L(q_{\phi_1}, q_{\phi_2}, \ldots, q_{\phi_K})$ ;

---

**Training Procedure** The training procedure consists of two phases:[1]

1. A single-decoder architecture (i.e. single agent) is trained from scratch, as shown in Figure 2 (left). This step is identical to the training process of Kwon et al. (2020).

2. The decoder trained in Phase 1 is cloned $K$ times to form a $K$-agent population. The whole model is trained using the approximated $J_{\text{poppy}}$ objective, as described in Algorithm 1 and illustrated in Figure 2 (right). Agents implicitly specialize on different types of problem instances during this phase.

Phase 1 enables training the large encoder without the computational overhead of a population. Moreover, we informally note that applying the Poppy objective directly to a population of untrained agents can be unstable. Randomly initialized agents are often ill-distributed, hence a single (or few) agent(s) dominate the performance across all instances. In this case, only the initially dominating agents receive a training signal, further widening the performance gap. Whilst directly training a population of untrained agents for population-level performance may be achievable with suitable modifications, we instead opt for the described pre-training approach as it is efficient and stable.

---

[1]Some of the experiments presented later were run using an obsolete 3-phase training procedure. The details are provided in Appendix A.2.

Figure 2: Phases of the training process. **Left (Phase 1)**: the encoder and the decoder are trained from scratch. **Right (Phase 2)**: the decoder is cloned $K$ times, and the whole model is trained using the Poppy training scheme (i.e. the gradient is only propagated through the decoder that yields the highest reward).

**Starting points**    Following Kwon et al. (2020), we generate multiple solutions for each instance $\rho$ by considering a set of $P \in [1, N]$ *starting points*, where $N$ is the number of instance variables. For example, a starting point in a TSP instance could be any of its cities. Therefore, across the different training phases, agents generate trajectories for (instance, starting point) pairs. The average reward across starting points is used as the REINFORCE baseline. By default, we consider each (instance, starting point) pair as a separate problem and, thus, train the best agent for each of them. We refer the reader to Appendix A.2 for details.

## 4    EXPERIMENTS

We evaluate Poppy on three CO problems: TSP, CVRP, and KP. To emphasize its generality, we use the same hyperparameters for each problem, taken from Kwon et al. (2020). We run Poppy for populations of 4, 8, 16, or 32 agents, which exhibit various time-performance tradeoffs.

**Training**    One training step corresponds to computing policy gradients over the same batch of 64 instances for each agent in the population. Training time varies with problem complexity and training phase. For instance, in TSP with 100 cities, Phase 1 takes 4.5M (5 days) steps, whereas Phase 2 takes 400k training steps and lasts 1-4 days depending on the population size. Our JAX-based implementation using environments from the Jumanji suite (Bonnet et al., 2022), along with problem instances to reproduce this work, are available at `https://anonymous.4open.science/r/poppy-6D20`. All experiments were run on a v3-8 TPU.

**Inference**    We greedily roll out each agent in the population, and use the augmentations proposed by Kwon et al. (2020) for TSP and CVRP. Additionally, to give a sense of the performance of Poppy with a larger time budget, we implement a simple sampling strategy. Given a population of $K$ agents, we first greedily rollout each of them on every starting point, and evenly distribute any remaining sampling budget across the most promising $K$ (agent, starting point) pairs for each instance.

**Baselines**    We compare Poppy against exact solvers, heuristics, and state-of-the-art ML methods. Some baseline performances taken from Fu et al. (2021), Xin et al. (2021) and Hottung et al. (2022) were obtained with different hardware (Nvidia GTX 1080 Ti, RTX 2080 Ti, and Tesla V100 GPUs, respectively) and framework (PyTorch); thus, for fairness, we mark these times with $*$ in our tables. As a comparison guideline, we informally note that these GPU inference times should be approximately divided by 2 to get the converted TPU time.

### 4.1    TRAVELING SALESMAN PROBLEM (TSP)

Given a set of $n$ cities, the goal in TSP is to visit every city and come back to the starting city while minimizing the total traveled distance.

**Setup** We use the architecture used by Kool et al. (2019) and Kwon et al. (2020) with slight modifications (see Appendix C). The testing instances are taken from Kool et al. (2019) for $n = 100$, and from Hottung et al. (2022) for $n \in \{125, 150\}$. The training is done on $n = 100$ instances. We compare Poppy to (i) the specialized supervised learning (SL) methods GCN-BS (Joshi et al., 2019), CVAE-Opt (Hottung et al., 2021), DPDP (Kool et al., 2021); (ii) the fast RL methods with limited sampling budgets POMO with greedy rollout, POMO with 16 stochastic rollouts (to match Poppy 16 runtime), POMO with an ensemble of 16 decoder heads trained in parallel (i.e. using the architecture of Poppy's Phase 2 but without specializing), MDAM (Xin et al., 2021) with greedy rollouts, and Att-GCRN+MCTS (Fu et al., 2021); and (iii) the slow RL methods (characterized by the inclusion of extensive search strategies) 2-Opt-DL (de O. da Costa et al., 2020), LIH (Wu et al., 2021), MDAM with beam search (beam size 50), POMO with 200 rollouts, and the fine-tuning approach EAS (Hottung et al., 2022). In addition to these ML-based approaches, we also compare to the exact solver Concorde (Applegate et al., 2006) and the heuristic solver LKH3 (Helsgaun, 2017).

**Results** Table 1 displays the average tour length, the optimality gap, and the total runtime for each test set. The best algorithm remains Concorde as it is a highly specialized TSP solver. Remarkably, Poppy 16 with greedy rollouts reaches the best performance across every category in just a few minutes, except for one case where DPDP performs better; however, DPDP tackles specifically routing problems and makes use of expert knowledge. With extra sampling, Poppy reaches a performance gap of $0.001\%$, which establishes a state-of-the-art for general ML-based approaches. Compared to DPDP, Poppy improves 0-shot performance, suggesting that it is more robust to distribution shifts. Att-GCRN+MCTS is known for being scalable to larger TSP instances than TSP100; however, it is outperformed by Poppy, showing that it trade offs performance for scale. Compared to the previous state-of-the-art fast RL method, POMO, Poppy 16 obtains strictly better performance in 81.67% of the TSP100 instances, and better or equal performance in 98.02%. Finally, we emphasize that specialization is crucial to achieve state-of-the-art performance: Poppy 16 outperforms POMO 16 (ensemble), which also trains 16 agents in parallel but without the $J_{\text{poppy}}$ objective (i.e. without specializing to serve as an ablation of our proposed objective).

Table 1: TSP results.

| | | **Inference** (10k instances) | | | **0-shot** (1k instances) | | | | | |
| | | $n = 100$ | | | $n = 125$ | | | $n = 150$ | | |
| | Method | Obj. | Gap | Time | Obj. | Gap | Time | Obj. | Gap | Time |
|---|---|---|---|---|---|---|---|---|---|---|
| | Concorde | 7.765 | 0.000% | 82M | 8.583 | 0.000% | 12M | 9.346 | 0.000% | 17M |
| | LKH3 | 7.765 | 0.000% | 8H | 8.583 | 0.000% | 73M | 9.346 | 0.000% | 99M |
| SL | GCN-BS | 7.87 | 1.39% | 40M* | - | - | - | - | - | - |
| | CVAE-Opt | - | 0.343% | 6D* | 8.646 | 0.736% | 21H* | 9.482 | 1.45% | 30H* |
| | DPDP | 7.765 | 0.004% | 2H* | 8.589 | 0.070% | 31M* | 9.434 | 0.94% | 44M* |
| RL (fast) | POMO | 7.774 | 0.13% | 37S | 8.605 | 0.26% | 6S | 9.393 | 0.50% | 10S |
| | POMO (16 samples) | 7.770 | 0.073% | 9M | 8.597 | 0.16% | 1M | 9.385 | 0.41% | 2M |
| | POMO 16 (ensemble) | 7.773 | 0.10% | 9M | 8.603 | 0.23% | 1M | 9.393 | 0.50% | 2M |
| | MDAM (greedy) | 7.93 | 2.19% | 36S* | - | - | - | - | - | - |
| | Att-GCRN+MCTS | - | 0.037% | 15M* | - | - | - | - | - | - |
| | **Poppy 4** | 7.767 | 0.029% | 2M | 8.590 | 0.079% | 23S | 9.364 | 0.19% | 38S |
| | **Poppy 8** | 7.766 | 0.015% | 5M | 8.587 | 0.046% | 45S | 9.360 | 0.14% | 1M |
| | **Poppy 16** | **7.765** | **0.008%** | 9M | **8.585** | **0.029%** | 1M | **9.356** | **0.10%** | 2M |
| RL (slow) | 2-Opt-DL | 7.83 | 0.87% | 41M* | - | - | - | - | - | - |
| | LIH | 7.87 | 1.42% | 2H* | - | - | - | - | - | - |
| | MDAM (beam search) | 7.79 | 0.38% | 44M* | - | - | - | - | - | - |
| | POMO (200 samples) | 7.769 | 0.056% | 2H | 8.594 | 0.13% | 20M | 9.376 | 0.31% | 32M |
| | EAS | 7.768 | 0.048% | 5H* | 8.591 | 0.091% | 49M* | 9.365 | 0.20% | 1H* |
| | **Poppy 16 (200 samples)** | 7.765 | **0.002%** | 2H | **8.584** | **0.009%** | 20M | **9.351** | **0.05%** | 32M |

**Analysis** Figure 3 helps understand the resulting behavior from using Poppy. The left plot shows that whilst the population-level performance improves with population size, the average performance of a random agent from the population gets worse. This shows a stronger specialization for larger populations, which $J_{\text{poppy}}$ appears to balance. Since Poppy produces an unsupervised specialization favoring the best agents, it is natural to consider how performance distributes across the population. In the right plot, we observe the performance is quite evenly distributed across the population of Poppy 16; hence, showing that the population has not collapsed to a few high-performing agents, and that Poppy benefits from the population size. Additional analyses are made in Appendix C.1.4.

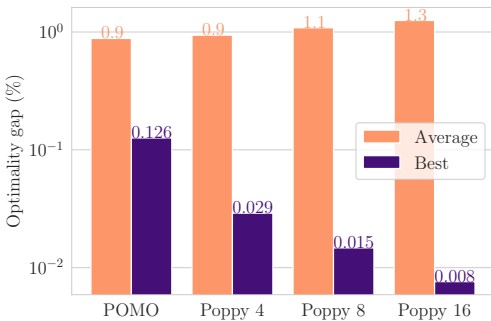 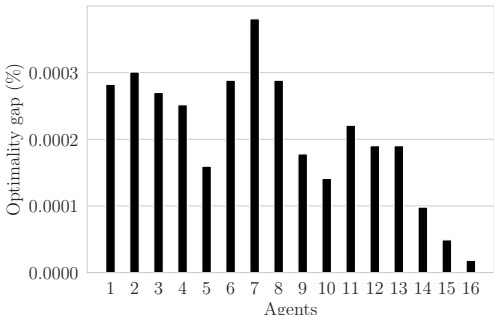

Figure 3: Analysis of Poppy on TSP100. **Left:** With $J_{\text{poppy}}$, the average performance gets worse as the population size increases, but the population-level performance improves. **Right:** Optimality gap loss suffered when removing any agent from the population using Poppy 16. Although some agents contribute more (e.g. 2, 7) and some less (e.g. 15, 16), the distribution is relatively even with all agents contributing, even though no explicit mechanism enforces this behavior.

## 4.2 CAPACITATED VEHICLE ROUTING PROBLEM (CVRP)

Given a vehicle with limited capacity departing from a depot node and a set of $n$ nodes with different demands, the goal is to find an optimal set of routes such that each node (except for the depot) is visited exactly once and has its demand covered. The vehicle's capacity diminishes by the demand of the visited node (which must be fully covered) and it is restored when the depot is visited.

**Setup** We use the test instances used by Kwon et al. (2020) for $n = 100$, and the sets from Hottung et al. (2022) to evaluate generalization to larger problems. We evaluate Poppy with populations of 4, 8 and 32 agents. We compare Poppy to the heuristic solver LKH3 (Helsgaun, 2017), taken as a reference to compute the gaps although its performances are not optimal. We also report results for the supervised ML methods CVAE-Opt (Hottung et al., 2021), DPDP (Kool et al., 2021), and RL methods NeuRewriter (Chen & Tian, 2019), NLNS (Hottung & Tierney, 2020), POMO (Kwon et al., 2020), MDAM (Xin et al., 2021), LIH (Wu et al., 2021), and EAS (Hottung et al., 2022). As for TSP, we evaluate POMO with greedy rollouts, 32 stochastic samples to match the runtime of our largest population, and 200 samples.

**Results** Table 2 shows Poppy has the best time-performance tradeoff among the fast approaches, e.g. Poppy 32 has the same runtime as POMO with 32 stochastic rollouts while dividing by 1.5 the optimal gap.[2] In addition, Poppy 32 is strictly better than POMO in 93.83% of the CVRP100 instances, and better or equal in 93.87%. Among the slow RL approaches, adding sampling to Poppy makes it on par with DPDP depending on the problem size, and it is only outperformed by the active search approach EAS, which gives large improvements on CVRP. However, active search (i) prevents parallelization (as it scales linearly with the number of samples) and (ii) could be added on top of Poppy instead of stochastic sampling to further boost performance, which we leave for future work.

## 4.3 0-1 KNAPSACK (KP)

We evaluate Poppy on KP to demonstrate its applicability beyond routing problems. Given a set of $n$ items with specific weights and values and a bag of limited capacity, the goal is to determine which items to add such that the total weight does not exceed the capacity and the value is maximal.

**Setup** We use the setting employed by Kwon et al. (2020): an action corresponds to putting an item in the bag, which is filled iteratively until no remaining item can be added. We evaluate Poppy on 3 population sizes against the optimal solution based on dynamic programming, a greedy heuristic, and POMO with and without sampling.

---

[2]A fine-grained comparison between POMO with stochastic sampling and Poppy is in Appendix 8.

Table 2: CVRP results.

| | Method | Inference (10k instances) $n = 100$ | | | 0-shot (1k instances) $n = 125$ | | | 0-shot (1k instances) $n = 150$ | | |
|---|---|---|---|---|---|---|---|---|---|---|
| | | Obj. | Gap | Time | Obj. | Gap | Time | Obj. | Gap | Time |
| | LKH3 | 15.65 | 0.000% | 6D | 17.50 | 0.000% | 19H | 19.22 | 0.000% | 20H |
| SL | CVAE-Opt | - | 1.36% | 11D* | 17.87 | 2.08% | 36H* | 19.84 | 3.24% | 46H* |
| | DPDP | 15.63 | −0.13% | 23H* | 17.51 | 0.07% | 3H* | 19.31 | 0.48% | 5H* |
| RL (fast) | NeuRewriter | 16.10 | - | 66M* | - | - | - | - | - | - |
| | NLNS | 15.99 | 2.23% | 62M* | 18.07 | 3.23% | 9M* | 19.96 | 3.86% | 12M* |
| | POMO | 15.76 | 0.76% | 2M | 17.68 | 1.02% | <1M | 19.58 | 1.85% | 1M |
| | POMO (32 samples) | 15.70 | 0.32% | 43M | 17.59 | 0.50% | 8M | 19.48 | 1.35% | 12M |
| | POMO 32 (ensemble) | 15.72 | 0.49% | 43M | 17.63 | 0.72% | 8M | 19.49 | 1.37% | 12M |
| | MDAM (greedy) | 16.40 | 4.86% | 45S* | - | - | - | - | - | - |
| | **Poppy 4** | 15.72 | 0.45% | 5M | 17.62 | 0.69% | 2M | 19.49 | 1.38% | 2M |
| | **Poppy 8** | 15.70 | 0.32% | 11M | 17.60 | 0.54% | 3M | 19.46 | 1.22% | 5M |
| | **Poppy 32** | **15.68** | **0.20%** | 43M | **17.57** | **0.40%** | 8M | **19.41** | **1.00%** | 12M |
| RL (slow) | LIH | 16.03 | 2.47% | 5H* | - | - | - | - | - | - |
| | MDAM (beam search) | 15.99 | 2.23% | 53M* | - | - | - | - | - | - |
| | POMO (200 samples) | 15.67 | 0.18% | 4H | 17.56 | 0.33% | 43M | 19.43 | 1.08% | 1H |
| | EAS | **15.62** | **-0.14%** | 8H* | **17.50** | **0.00%** | 80M* | 19.36 | 0.72% | 2H* |
| | **Poppy 32 (200 samples)** | 15.63 | −0.10% | 4H | 17.51 | 0.02% | 42M | **19.34** | **0.61%** | 1H |

**Results**  Table 3 shows that Poppy leads to improved performance with a population of 16 agents, dividing the optimality gap with respect to POMO by 45 and 12 on KP100 and KP200 respectively, and by 12 and 2 with respect to POMO with 16 stochastic samples for the exact same runtime. Poppy 16 is strictly better than POMO in 34.30% of the KP100 instances, and better in 99.95%.

Table 3: KP results.

| Method | Testing (10k instances) $n = 100$ | | | Testing (1k instances) $n = 200$ | | |
|---|---|---|---|---|---|---|
| | Obj. | Gap | Time | Obj. | Gap | Time |
| Optimal | 40.437 | - | | 57.729 | - | |
| Greedy | 40.387 | 0.1250% | | 57.672 | 0.0986% | |
| POMO | 40.428 | 0.0224% | 8S | 57.718 | 0.0191% | 4S |
| POMO (16 samples) | 40.435 | 0.0060% | 2M | 57.727 | 0.0032% | 1M |
| POMO 16 (ensemble) | 40.429 | 0.021% | 2M | 57.719 | 0.0170% | 1M |
| **Poppy 4** | 40.434 | 0.0081% | 33S | 57.723 | 0.0099% | 16S |
| **Poppy 8** | 40.436 | 0.0032% | 1M | 57.726 | 0.0058% | 33S |
| **Poppy 16** | **40.437** | **0.0005%** | 2M | **57.728** | **0.0015%** | 1M |

## 5 CONCLUSIONS

Poppy is a population-based RL method for CO problems. It uses an RL objective that incurs agent specialization with the purpose of maximizing population-level performance. Crucially, Poppy does not rely on handcrafted notions of diversity to enforce specialization. We show that Poppy achieves state-of-the-art performance on three popular NP-hard problems: TSP, CVRP, and KP.

This work raises several questions. First, we have experimented on populations of at most 32 agents; therefore, it is unclear what the consequences of training larger populations are. We hypothesize that the population performance would eventually collapse, leaving agents with null contributions behind. Exploring this direction, including possible strategies to prevent such collapses, is an interesting direction for future work. Second, we recall that the motivation behind Poppy was dealing with problems where predicting optimal actions from observations is too difficult to be solved reliably by a single agent. We believe that such settings may not be strictly limited to canonical CO problems, and that population-based approaches offer a promising direction for many challenging RL applications. In this direction, we hope that approaches (such as Poppy) that alleviate the need for handcrafted behavioral markers, whilst still realizing performant diversity, could broaden the range of applications of population-based RL.

REPRODUCIBILITY STATEMENT

We provide in Appendix A.2 the detailed pseudocode of our method. Model architectures as well as the training hyperparameters are given in Appendices C.1.2, C.2.2 and C.4 for TSP, CVRP and KP respectively. Source code and test instances are available at `https://anonymous.4open.science/r/poppy-6D20`.

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

# A ADDITIONAL DETAILS ON POPPY

## A.1 NUMBER OF PARAMETERS

Table 4 shows the total number of parameters of our models as a function of the population size. Since the decoder represents less than $10\%$ of the parameters, scaling the population size can be done efficiently. For instance, a population of 16 agents roughly doubles the model size.

Table 4: Number of model parameters for different population sizes.

|  | Encoder | Decoder | Population size | | | | |
| --- | --- | --- | --- | --- | --- | --- | --- |
|  |  |  | 1 | 4 | 8 | 16 | 32 |
| Parameters | 1,190,016 | 98,816 | 1,288,832 | 1,585,280 | 1,980,544 | 2,771,072 | 4,352,128 |
| Extra parameters | - | - | 0% | 23% | 54% | 115% | 238% |

## A.2 TRAINING DETAILS

In Section 3.2 (see "Training Procedure"), we described that Poppy consists of two phases. In a nutshell, the first phase consists of training our model in a single-agent setting (i.e., an encoder-decoder model with a single decoder head), whereas the second phase consists of keeping the encoder and cloning the previously trained decoder $K$ times (where $K$ is the number of agents) and specialize them using the $J_{\text{poppy}}$ objective. Algorithm 2 shows the low-level implementation details of the training of the population (i.e., Phase 2) omitted in Algorithm 1 for simplicity; namely, given $K$ agents and $P$ starting points, $P \times K$ trajectories are rolled out for each instance, among which only $P$ are effectively used for training.

For historical reasons, we describe an alternative procedure which instead uses of three phases to achieve the same results. The experimental results for CVRP and KP (Sections 4.2 and 4.3) were obtained using this three-phase training procedure; however, with both approaches leading to equivalent performance, retraining all models with the two-phase procedure is not necessary. In what follows, we detail these phases:

1. Equivalent to the current Phase 1.

2. The trained decoder is discarded and a population of decoder heads is randomly initialized. With the parameters of the shared encoder frozen, the decoders are trained in parallel using the same training instances.

3. The encoder is unfrozen and trained jointly with the population decoders as described in the current Phase 2 (Section 3.2).

---

**Algorithm 2:** Poppy training with starting points

---

**input:** problem distribution $\mathcal{D}$, number of starting points per instance $P$, number of agents $K$,
batch size $B$, number of training steps $H$, a pretrained encoder $h_\psi$ and a set of $K$
decoders $q_{\phi_1}, q_{\phi_2}, \ldots, q_{\phi_K}$ ;

**for** *step 1 to H* **do**

    $\rho_i \leftarrow \texttt{Sample}(\mathcal{D}) \, \forall i \in 1, \ldots, B$ ;

    $\alpha_{i,1}, \ldots, \alpha_{i,P} \leftarrow \texttt{SelectStartPoints}(\rho_i, P) \, \forall i \in 1, \ldots, B$ ;

    $\tau_{i,p}^k \leftarrow \texttt{Rollout}(\rho_i, \alpha_{i,p}, h_\psi, q_{\phi_k}) \, \forall i \in 1, \ldots, B, \forall p \in 1, \ldots, P, \forall k \in 1, \ldots, K$ ;

    $b_i^k \leftarrow \frac{1}{P} \sum_p R(\tau_{i,p}^k)$ ;

    /* Select the best agent per (instance, starting point).  */

    $k_{i,p}^* \leftarrow \arg\max_{k \leq K} R(\tau_{i,p}^k) \, \forall i \in 1, \ldots, B, \forall p \in 1, \ldots, P$ ;

    /* Propagate the gradients through these only.  */

    $\nabla L(h_\psi, q_{\phi_1}, q_{\phi_2}, \ldots, q_{\phi_K}) \leftarrow -\frac{1}{BP} \sum_{i,p} (R(\tau_{i,p}^{k_{i,p}^*}) - b_i^{k_{i,p}^*}) \nabla \log p_{\psi, \phi_{k_{i,p}^*}}(\tau_{i,p}^{k_{i,p}^*})$ ;

    $(h_\psi, q_{\phi_1}, q_{\phi_2}, \ldots, q_{\phi_K}) \leftarrow (h_\psi, q_{\phi_1}, q_{\phi_2}, \ldots, q_{\phi_K}) - \alpha \nabla L(h_\psi, q_{\phi_1}, q_{\phi_2}, \ldots, q_{\phi_K})$ ;

---

The crucial difference that motivates the use of an additional phase is the fact that after Phase 1, the decoder heads are not cloned but randomly initialized. However, the purpose is exactly the same: have generally capable agents before specializing in order to avoid that only a few dominating agents are trained. We have opted for cloning the decoder from Phase 1 instead of training a set of non-specialized decoders in parallel since: (i) there are less phases, and (ii) it is less computationally demanding (i.e. we do not need to spend training time unnecessarily). Empirically, the training of the population is performed twice as fast with the two-phase procedure in TSP100.

## B  MATHEMATICAL ELEMENTS

### B.1  BIAS IN THE GRADIENT ESTIMATION OF THE OBJECTIVE

We recall that the instance distribution is $\mathcal{D}$, and that for the parameters $\bar{\theta} = \{\theta_1, \theta_2, \ldots, \theta_K\}$ of a population of $K$ agents, $J_{\text{poppy}}$ is defined as:

$$J_{\text{poppy}} = \mathbb{E}_{\rho \sim \mathcal{D}} \left[ \mathbb{E}_{\tau \sim \pi_{\theta_\rho^*}} R(\tau) \right] \text{ with } \theta_\rho^* = \arg\max_{\theta_i \in \bar{\theta}} \mathbb{E}_{\tau \sim \pi_{\theta_i}, \rho} R(\tau).$$

The purpose of this section is to quantify the bias between the gradient of $J_{\text{poppy}}$, and the empirical estimate computed from Alg. 1, that we denote $\widetilde{\nabla}_{\bar{\theta}} J_{\text{poppy}}$.

$$
\begin{aligned}
\nabla_{\bar{\theta}} J_{\text{poppy}} &= \mathbb{E}_{\rho \sim \mathcal{D}} \left[ \nabla_{\bar{\theta}} \mathbb{E}_{\tau \sim \pi_{\theta_\rho^*}} R(\tau) \right] \\
&= \mathbb{E}_{\rho \sim \mathcal{D}} \left[ \nabla_{\bar{\theta}} \mathbb{E}_{\tau_1 \sim \pi_{\theta_1}, \tau_2 \sim \pi_{\theta_2}, \ldots, \tau_K \sim \pi_{\theta_K}} \sum_{k=1}^{K} \mathbb{1}_{\pi_{\theta_k} = \pi_{\theta_\rho^*}} R(\tau_k) \right] \\
&= \mathbb{E}_{\rho \sim \mathcal{D}} \left[ \sum_{k=1}^{K} \mathbb{1}_{\pi_{\theta_k} = \pi_{\theta_\rho^*}} \nabla_{\bar{\theta}} \mathbb{E}_{\tau_1 \sim \pi_{\theta_1}, \tau_2 \sim \pi_{\theta_2}, \ldots, \tau_K \sim \pi_{\theta_K}} R(\tau_k) \right] \\
&= \mathbb{E}_{\rho \sim \mathcal{D}} \left[ \sum_{k=1}^{K} \mathbb{1}_{\pi_{\theta_k} = \pi_{\theta_\rho^*}} \nabla_{\theta_k} \mathbb{E}_{\tau_k \sim \pi_{\theta_k}} R(\tau_k) \right] \\
&= \mathbb{E}_{\rho \sim \mathcal{D}} \left[ \sum_{k=1}^{K} \mathbb{1}_{\pi_{\theta_k} = \pi_{\theta_\rho^*}} \mathbb{E}_{\tau_k \sim \pi_{\theta_k}} \nabla_{\theta_k} R(\tau_k) \log p_{\pi_{\theta_k}}(\tau_k) \right] \\
&= \mathbb{E}_{\rho \sim \mathcal{D}} \left[ \sum_{k=1}^{K} \mathbb{E}_{\tau_k \sim \pi_{\theta_k}} \mathbb{1}_{\pi_{\theta_k} = \pi_{\theta_\rho^*}} \nabla_{\theta_k} R(\tau_k) \log p_{\pi_{\theta_k}}(\tau_k) \right]
\end{aligned}
$$

$$= \mathbb{E}_{\rho \sim \mathcal{D}} \left[ \sum_{k=1}^{K} \mathbb{E}_{\tau_1 \sim \pi_{\theta_1}, \tau_2 \sim \pi_{\theta_2}, \ldots, \tau_K \sim \pi_{\theta_K}} \mathbb{1}_{\pi_{\theta_k} = \pi_{\theta_\rho^*}} \nabla_{\theta_k} R(\tau_k) \log p_{\pi_{\theta_k}}(\tau_k) \right].$$

We now derive an expression of $\widetilde{\nabla}_{\bar{\theta}} J_{\text{poppy}}$. For a given instance $\rho$, and $\tau_1, \ldots, \tau_K$ random trajectories sampled from the $K$ agents in the population, we denote $\tau_\rho^* = \arg\max_{\tau_k} [R(\tau_k)]$ the best trajectory in the population. The gradient computed with Alg. 1 is:

$$\widetilde{\nabla}_{\bar{\theta}} J_{\text{poppy}}(\rho, \tau_1, \ldots, \tau_K) = \sum_{k=1}^{K} \mathbb{1}_{\tau_k = \tau_\rho^*} \nabla_{\theta_k} R(\tau_k) \log p_{\pi_{\theta_k}}(\tau_k).$$

Therefore, we can derive the expected gradient:

$$\mathbb{E} \left[ \widetilde{\nabla}_{\bar{\theta}} J_{\text{poppy}} \right] = \mathbb{E}_{\rho \sim \mathcal{D}} \mathbb{E}_{\tau_1 \sim \pi_{\theta_1}, \tau_2 \sim \pi_{\theta_2}, \ldots, \tau_K \sim \pi_{\theta_K}} \sum_{k=1}^{K} \mathbb{1}_{\tau_k = \tau_\rho^*} \nabla_{\theta_k} R(\tau_k) \log p_{\pi_{\theta_k}}(\tau_k)$$

$$= \mathbb{E}_{\rho \sim \mathcal{D}} \left[ \sum_{k=1}^{K} \mathbb{E}_{\tau_1 \sim \pi_{\theta_1}, \tau_2 \sim \pi_{\theta_2}, \ldots, \tau_K \sim \pi_{\theta_K}} \mathbb{1}_{\tau_k = \tau_\rho^*} \nabla_{\theta_k} R(\tau_k) \log p_{\pi_{\theta_k}}(\tau_k) \right].$$

The bias can thus be written as:

$$\mathbb{E} \left[ \widetilde{\nabla}_{\bar{\theta}} J_{\text{poppy}} \right] - \nabla_{\bar{\theta}} J_{\text{poppy}} =$$

$$\mathbb{E}_{\rho \sim \mathcal{D}} \left[ \sum_{k=1}^{K} \mathbb{E}_{\tau_1 \sim \pi_{\theta_1}, \tau_2 \sim \pi_{\theta_2}, \ldots, \tau_K \sim \pi_{\theta_K}} (\mathbb{1}_{\tau_k = \tau_\rho^*} - \mathbb{1}_{\pi_{\theta_k} = \pi_{\theta_\rho^*}}) \nabla_{\theta_k} R(\tau_k) \log p_{\pi_{\theta_k}}(\tau_k) \right].$$

Thus, denoting by $\mathcal{M}$ an upper bound of the gradient norm (e.g., using clipped gradient),

$$\left\| \mathbb{E} \left[ \widetilde{\nabla}_{\bar{\theta}} J_{\text{poppy}} \right] - \nabla_{\bar{\theta}} J_{\text{poppy}} \right\| \leq \mathcal{M} \mathbb{E}_{\rho \sim \mathcal{D}} \left[ \sum_{k=1}^{K} \mathbb{E}_{\tau_1 \sim \pi_{\theta_1}, \tau_2 \sim \pi_{\theta_2}, \ldots, \tau_K \sim \pi_{\theta_K}} \| \mathbb{1}_{\tau_k = \tau_\rho^*} - \mathbb{1}_{\pi_{\theta_k} = \pi_{\theta_\rho^*}} \| \right],$$

$$\left\| \mathbb{E} \left[ \widetilde{\nabla}_{\bar{\theta}} J_{\text{poppy}} \right] - \nabla_{\bar{\theta}} J_{\text{poppy}} \right\| \leq \mathcal{M} \mathbb{E}_{\rho \sim \mathcal{D}} \mathbb{E}_{\tau_1 \sim \pi_{\theta_1}, \tau_2 \sim \pi_{\theta_2}, \ldots, \tau_K \sim \pi_{\theta_K}} \underbrace{\sum_{k=1}^{K} \| \mathbb{1}_{\tau_k = \tau_\rho^*} - \mathbb{1}_{\pi_{\theta_k} = \pi_{\theta_\rho^*}} \|}_{2 \text{ if } \tau_\rho^* \text{ was not sampled from } \pi_{\theta_\rho^*}, \text{ else } 0},$$

$$\left\| \mathbb{E} \left[ \widetilde{\nabla}_{\bar{\theta}} J_{\text{poppy}} \right] - \nabla_{\bar{\theta}} J_{\text{poppy}} \right\| \leq 2\mathcal{M} \mathbb{E}_{\rho \sim \mathcal{D}} \mathbb{E}_{\tau_1 \sim \pi_{\theta_1}, \tau_2 \sim \pi_{\theta_2}, \ldots, \tau_K \sim \pi_{\theta_K}} (1 - \mathbb{1}_{\tau_\rho^* \sim \pi_{\theta_\rho^*}}).$$

Denoting by $k_\rho^*$ the index of $\theta_\rho^*$, corresponding to the best agent in the population for the instance $\rho$, we thus have:

$$\left\| \mathbb{E} \left[ \widetilde{\nabla}_{\bar{\theta}} J_{\text{poppy}} \right] - \nabla_{\bar{\theta}} J_{\text{poppy}} \right\| \leq 2\mathcal{M} \cdot p \left( \tau_\rho^* \neq \tau_{k_\rho^*} \right).$$

Controlling the bias amounts to controlling $p(\tau_\rho^* \neq \tau_{k_\rho^*})$. Intuitively, it is the probability that the best trajectory sampled by rolling out the whole population does not belong the best agent. It is to be noted that the bias is null in the limit of deterministic agents, or if agents are specialized (meaning that the best agent in expectancy on one instance $\rho$ consistently outperforms the other agents this instance). We provide empirical insights on $p(\tau_\rho^* \neq \tau_{k_\rho^*})$ in the next section.

## B.2 EMPIRICAL BIAS ESTIMATION

To have a sense of $p \left( \tau_\rho^* \neq \tau_{k_\rho^*} \right)$ in our setting, we computed this quantity for several early check-points for TSP, obtained after few minutes of training. Interestingly, we can see on Fig. 4 that

$p\left(\tau_\rho^* \neq \tau_{k_\rho^*}\right)$ (i.e the probability that the best trajectory among one sampling of the population does not belong to the best agent on average) decreases quickly at the very beginning of the training, which we interpret as specialization. A consequence is that our gradient estimation of $J_{\text{poppy}}$ is less biased as the training progresses. Another observation is that this probability increases with the population size, which is logical as the number of competitors to the best agent is greater. Although the bias is thus larger for large populations, the empirical performances seem not to be affected, as seen in Table 1. These results raise a question, which we leave for future work: would techniques to alleviate the gradient bias improve the performances further, or are there other mechanisms at play which explain the good performance of large populations?

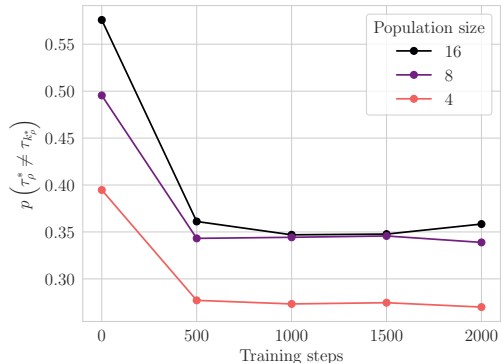

Figure 4: $p\left(\tau_\rho^* \neq \tau_{k_\rho^*}\right)$ computed on 20 seeds for various early checkpoints on TSP.

## C  PROBLEMS

We here describe the details of the CO problems we have used to evaluate Poppy, including instance generation and training details (e.g. architecture, hyperparameters). In the case of TSP and CVRP, we show some example solutions obtained by a population of agents. Besides, we thoroughly analyze the performance of the populations in TSP.

### C.1  TRAVELING SALESMAN PROBLEM (TSP)

#### C.1.1  INSTANCE GENERATION

The $n$ cities that constitute each problem's instance are uniformly sampled from $[0, 1]^2$.

#### C.1.2  TRAINING DETAILS

**Architecture**   We use the same model as Kool et al. (2019) and Kwon et al. (2020) except for the batch-normalization layers, which are replaced by layer-normalization to ease parallel batch processing. We invert the mask used in the decoder computations (i.e., masking the available cities instead of the unavailable ones) after having experimentally observed faster convergence rates.

**Hyperparameters**   To match the setting used by Kwon et al. (2020), we use the Adam optimizer (Kingma & Ba, 2015) with a learning rate $\mu = 10^{-4}$, and a $L_2$ penalization of $10^{-6}$. The encoder is composed of 6 multi-head attention layers with 8 heads each. The dimension of the keys, queries and values is 16. Each attention layer is composed of a feed-forward layer of size 512, and the final node embeddings have 128 dimensions. The decoders are composed of 1 multi-head attention layer with 8 heads and 16-dimensional key, query and value.

The number of starting points $P$ is 20 for each instance. We determined this value after performing a grid-search based on the first training steps with $P \in \{20, 50, 100\}$.

### C.1.3 EXAMPLE SOLUTIONS

Figure 6 shows some trajectories obtained from a 16-agent population on TSP100. Even though they look similar, small decisions differ between agents, thus frequently leading to different solutions. Interestingly, some agents (especially 6 and 11) give very poor trajectories. We hypothesize that it is a consequence of specializing since agents have no incentive to provide a good solution if another agent is already better on this instance.

### C.1.4 POPULATION ANALYSIS

Figure 5 shows some additional information about individual agent performances. In the left figure, we observe that each agent gives on average the best solution for 35% of the instances, and that for around 2.5% it gives the unique best solution across the population. These numbers are evenly distributed, which shows that every agent contributes to the whole population performance. Furthermore, the best performance is reached by a single agent in around 25% of the cases, as shown in the bottom figure. On the right is displayed the performance of several sub-populations of agents for Poppy 4, 8 and 16. Unsurprisingly, any fixed size sub-population is better when sampled from smaller populations: Poppy 16 needs 4 agents to recover the performance of Poppy 4, and 8 agents to recover the performance of Poppy 8 for example. This highlights the fact that agents have learned complementary behaviors which might be sub-optimal if part of the total population is missing.

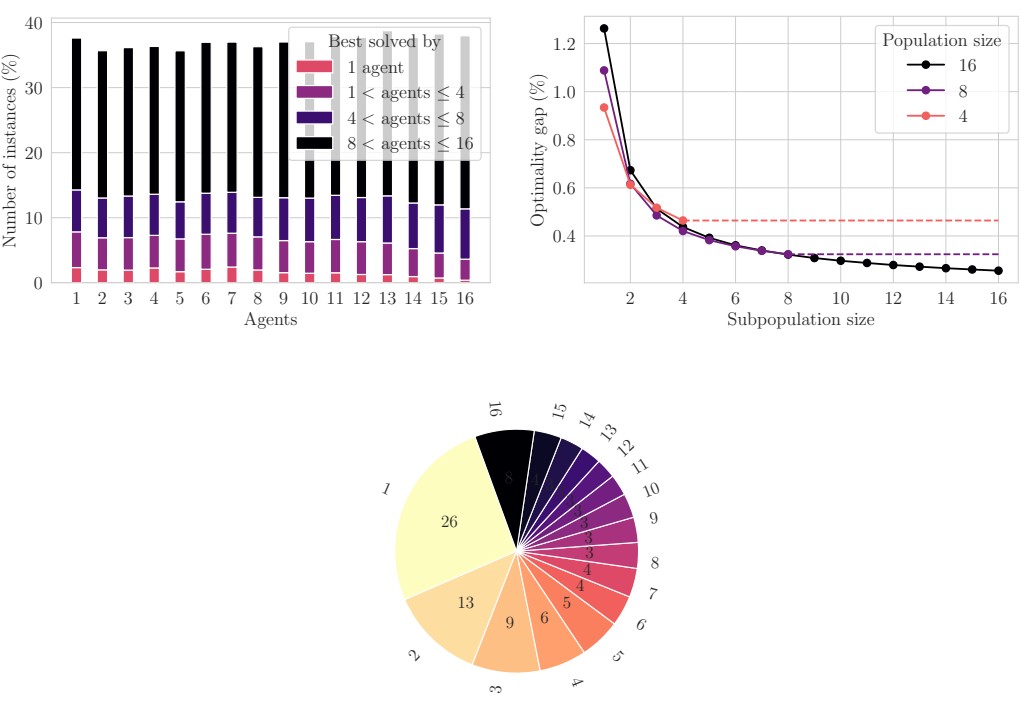

Figure 5: **Left**: Proportion of instances that each agent solves best among the population for Poppy 16 on TSP100. Colors indicate the number of agents in the population giving the same solution for these sets of instances. **Right**: The mean performance of 1,000 randomly drawn sub-populations for Poppy 1, 4, 8 and 16. **Bottom**: Proportion of test instances where any number of agents reach the exact same best solution. A 26% of the instances are best solved by a single agent, and almost a 50% by three agents or less.

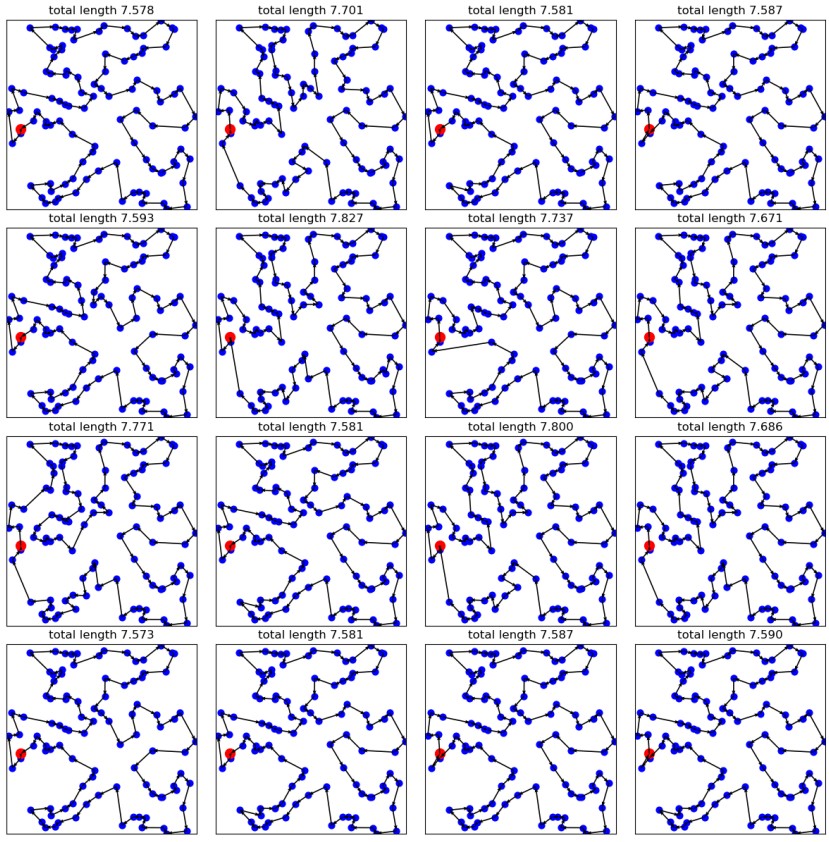

Figure 6: Example TSP trajectories given by Poppy for a 16-agent population from one starting point (red).

## C.2 CAPACITATED VEHICLE ROUTING PROBLEM (CVRP)

### C.2.1 INSTANCE GENERATION

The locations of the $n$ customer nodes and the depot are uniformly sampled in $[0, 1]^2$. The demands are uniformly sampled from the discrete set $\left\{\frac{1}{D}, \frac{2}{D}, \ldots, \frac{9}{D}\right\}$ where $D = 50$ for CVRP100, $D = 55$ for CVRP125, and $D = 60$ for CVRP150. The maximum vehicle capacity is 1. The deliveries cannot be split: each customer node is visited once, and its whole demand is taken off the vehicle's remaining capacity.

### C.2.2 TRAINING DETAILS

**Architecture** We use the same model as in TSP. However, unlike TSP, the mask is not inverted; besides, it does not only prevent the agent from revisiting previous customer nodes, but also from visiting the depot if it is the current location, and any customer node whose demand is higher than the current capacity.

**Hyperparameters** We use the same hyperparameters as in TSP except for the number of starting points $P$ per instance used during training, which we set to 50 after performing a grid-search with $P \in \{20, 50, 100\}$.

### C.2.3 EXAMPLE SOLUTIONS

Figure 7 shows some trajectories obtained by 16 agents from a 32-agent population on CVRP100. Unlike TSP, the agent/vehicle performs several tours starting and finishing in the depot.

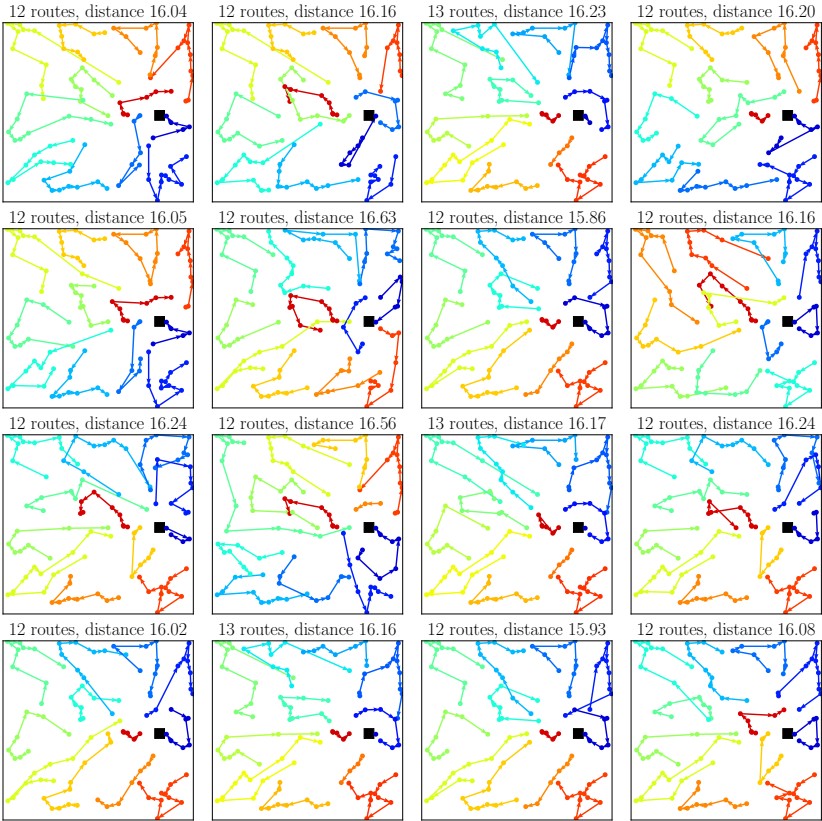

Figure 7: Example CVRP trajectories given by Poppy for 16 agents from a 32-agent population. The depot is displayed as a black square. The edges from/to the depot are omitted for clarity.

### C.3 0-1 KNAPSACK (KP)

### C.3.1 INSTANCE GENERATION

Item values and weights are uniformly sampled in $[0, 1]$. The bag capacity is fixed to 25.

### C.4 TRAINING DETAILS

**Architecture** We use the same model as in TSP. However, the mask used when decoding is not inverted, and the items that do not fit in the bag are masked together with the items taken so far.

**Hyperparameters**   We use the same hyperparameters as in TSP except for the number of starting points $P$ used during training, which we set to 100 after performing a grid-search with $P \in \{20, 50, 100\}$.

## D   TIME-PERFORMANCE TRADEOFF

We present on Figure 8 a comparison of the time-performance Pareto front between Poppy and POMO as we vary respectively the population size and the amount of stochastic sampling. Poppy consistently provides better performance for a fixed number of trajectories. Strikingly, in some environments like TSP100, TSP125, TSP150, CVRP150 and KP100, matching Poppy's performance by increasing the number of stochastic samples does not appear tractable.

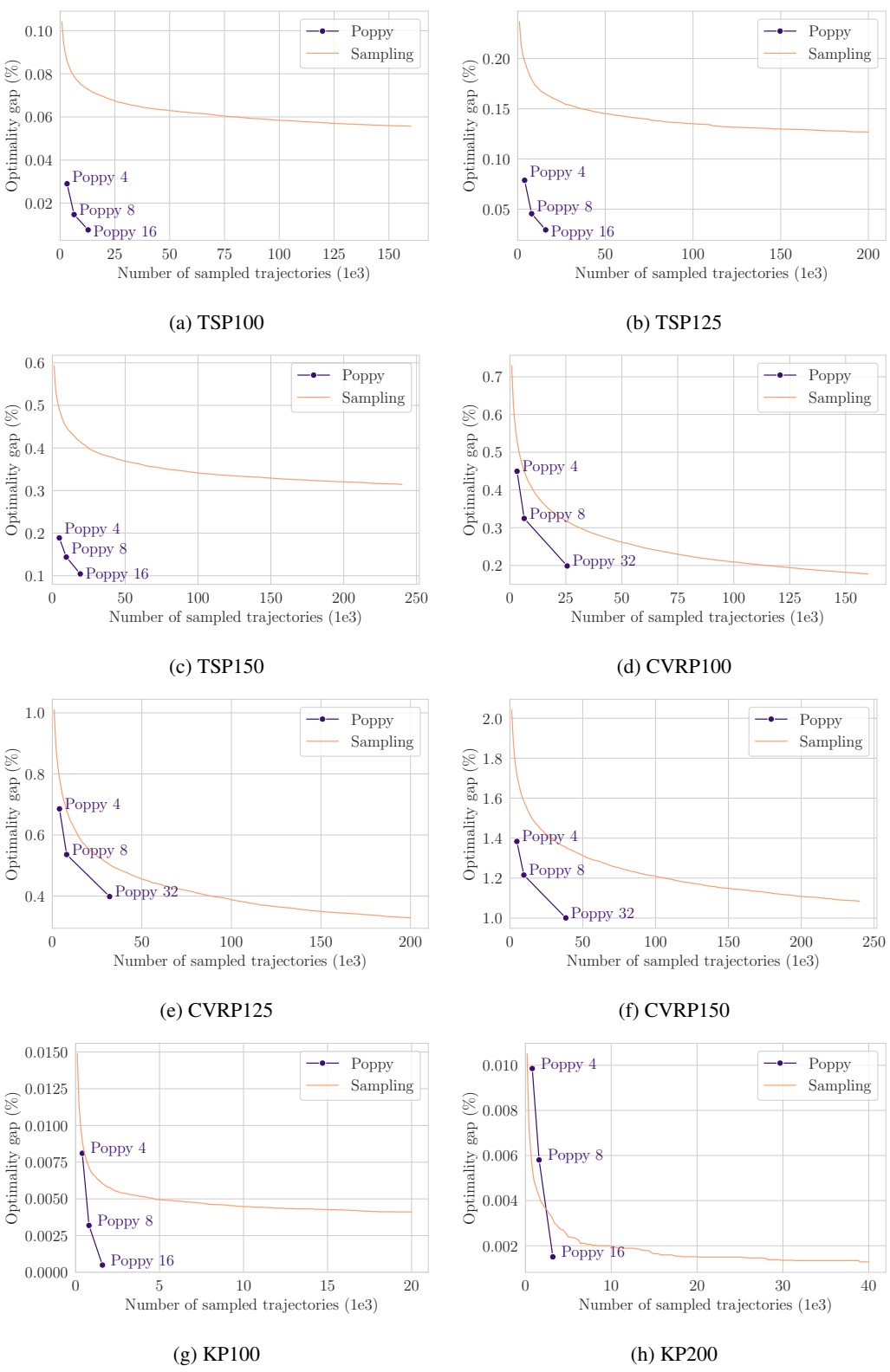

Figure 8: Comparison of the time-performance Pareto front of Poppy and POMO, for each problem used in the paper. The x-axis is the number of trajectories sampled per test instance, while the y-axis is the gap with the optimal solution for TSP and KP, and the gap with LKH3 for CVRP.

