# OpenReview forum: "Population-Based Reinforcement Learning for Combinatorial Optimization Problems"
_ICLR.cc/2023/Conference — Submitted to ICLR 2023_

### Official Review · Reviewer_rRMu · 2022-10-20

**Confidence:** 3
**Correctness:** 3
**Technical Novelty And Significance:** 2
**Empirical Novelty And Significance:** 2
**Recommendation:** 5

**Clarity, Quality, Novelty And Reproducibility:**

**Clarity and Quality:** My main concern is with the theoretical motivations. Please see above in "Weaknesses". Furthermore, there are some presentation issues, e.g. Figure 3 is not described very well / the plots are confusing and unclear of their purpose. Otherwise the experimental results appear to be solid.

**Novelty and Originality:** The proposed idea has **potential** to be quite novel, if the theoretical components are analyzed better. However, currently it is presented as a somewhat ad-hoc experimental method.

**Strength And Weaknesses:**

# Strengths
* Extensive experimental results demonstrate that the method empirically is solid. A brief read over the Appendix also contains relevant ablation studies, such as checking the diversity of the population of policies.
* I applaud the paper for providing Figures 1 and 2, which makes the paper more engaging to read, and also help explain the encoder/decoder method much better.


# Weaknesses
* The theoretical motivations are quite subtle and I have quite a few questions, which if answered, could help make the paper much stronger in its theoretical/fundamental contributions:
    * How is the framework similar/different from e.g. describing evaluation objective as a maximum of $K$ sampled trajectories from a (potentially very expressive) policy $\pi_{\theta_{meta}}$? In the $K=2$ case, $\pi_{\theta_{meta}}$ for example, could be a very bimodal policy which can also produce diverse behaviors from its two modes.
    * Is Algorithm 1 an unbiased policy gradient estimator of $J_{poppy}$?
    * The method has no explicit objective/regularizer to maximize behavioral diversity, but instead implicitly assumes that this occurs through the "argmax" training in Algorithm 1. Is there a way to theoretically capture/analyze this "implicit regularization"? Currently it seems that the behavioral diversity isn't theoretically guaranteed, but rather occurs I suspect mainly if the beginning of training allows each agent to "latch on" to their own specific strategy.
    * Algorithm 1 is written somewhat specifically for this particular case involving a shared encoder and multiple decoders. Can it be re-written as a general RL method?

**Summary Of The Paper:**

In summary, from top to bottom, the paper:
* Attempts to explain why a population of agents may be more useful than a single agent, under a theoretical formulation of argmax(multiple agent outputs).
* Presents a lower bound objective which leads to a variant of policy gradient which involves only updating the policy which generated the best trajectory per problem instance.
* Sets up the architecture as a shared encoder but with multiple decoders to denote multiple policies.
* Demonstrates competitive experimental performance against classic and deep-learning based baselines in the literature, especially normalizing for wall-clock time.

**Summary Of The Review:**

The experimental results are fine. However, I would really like to see more theoretical analysis of the proposed method, which would significantly increase the impact of the paper and also broaden its scope. Otherwise, currently from a devil's advocate point of view, the proposed method may be considered somewhat of a "hack" currently.

---

> ### Author Response · Authors · 2022-11-14
> **Authors' response to Reviewer rRMu [Part 2/2]**
>
> > “Is Algorithm 1 an unbiased policy gradient estimator of $J_{poppy}$?”
>
> This is an interesting question to which we have added a detailed analysis in Appendix B. In brief, the answer is determined by whether the index of the best agent is well estimated by the one-shot estimate used in our training process. In general, the gradient is therefore biased, however it becomes unbiased in the limits of either (1) deterministic policies (i.e. the best trajectory amongst those generated will be guaranteed to come from the same policy every time) or (2) perfectly specialized agents (i.e. the stochastic trajectories sampled from the best agents are always better than trajectories of the other agents). We have emphasized in the paper that our algorithm is an approximate optimization of $J_{poppy}$. Empirically, by observing early training checkpoints, we find that our estimation of the best agent improves as training progresses, and is correct roughly 2/3 of the time after a few training iterations.
>
> > “The method has no explicit objective/regularizer to maximize behavioral diversity, but instead implicitly assumes that this occurs through the "argmax" training in Algorithm 1. Is there a way to theoretically capture/analyze this "implicit regularization"? Currently it seems that the behavioral diversity isn't theoretically guaranteed…”
>
> Indeed diversity is not guaranteed (as we now state in “Population-Based Training Objective” in Sec. 3.2); we propose an objective for which diversity naturally emerges over training in the pursuit of its maximization, rather than explicitly adding some driver of diversity (e.g. KL-divergence of policies). Indeed, we consider the lack of explicit behavioral regularization to be a significant strength of our approach, as it is not clear what performant diversity should look like for NP-hard combinatorial problems nor how best to balance “RL” and explicit “diversity” objectives. However, our approach is reliable and robust. Note that our objective (“maximize the performance of the best agent in the population on each problem”) would only be maximized by a non-diverse population if a single policy is the best obtainable strategy on any/all subsets of problems from the training distribution. We motivate why this is highly unlikely to be the case in our “Motivating Example” (Sec 3.1) and further back this up by empirically observing diversity has emerged. Whilst the reviewer is correct that agents “latch on” to specific strategies, we would note that our new training procedure – where the entire population is initialized to the same pre-trained policy – demonstrates that our algorithm still causes performant diversity to emerge even from initially identical agents.
>
> > “Algorithm 1 is written somewhat specifically for this particular case involving a shared encoder and multiple decoders. Can it be re-written as a general RL method?”
>
> We agree that it would lead to a clearer understanding, as the training method is not tied to this particular architecture! We have updated Algorithm 1, such that it is independent of the model architecture (whilst still providing the specific algorithm for our case in Appendix A.2 - Algorithm 2).
>
> > “Figure 3 is not described very well / the plots are confusing and unclear of their purpose.”
>
> We have simplified Figure 3. First, we have removed the top two plots since Poppy is now presented as a two-phase process. These plots showed the changes in population and average performance when switching from Phase 2 to Phase 3, but these no longer happen given that Phase 2 (training a set of randomly initialized decoders in parallel without specializing) does not occur. Second, the old bottom right plot is now a bar chart showing the average and population performances at the end of the training instead of showing the rather confusing learning curves.

---

> ### Author Response · Authors · 2022-11-14
> **Authors' response to Reviewer rRMu [Part 1/2]**
>
> Thank you for your consideration and insightful comments. Whilst we provide point-by-point responses to your questions below, we would like to address your overall comments that, whilst experimentally sound and with potential novelty, you consider our weakness to be (1) the theoretical aspects of our work and (2) “hacky” nature of the implementation.
>
> (1) We consider the primary contribution of our work to be the pursuit and realization of specialized populations of policies for intractable combinatorial problems; along with the significant SOTA results achieved. Our intent was not to provide rigorous theoretical/convergence guarantees for training a population of agents, but rather to give a theoretical justification of our intuition for the Poppy objective and the substantial empirical study of it we perform. We accept that the original text should make this distinction clearer, and so have (i) removed “theoretically grounded” from the abstract, (ii) provided extended analysis of the Poppy objective’s gradient (details below), and (iii) made clear that Poppy does not provide any strict diversity or performance guarantee (see “Population-Based Training Objective” in Sec. 3.2). We believe these modifications strike the right balance between providing theoretical analysis of Poppy whilst not detracting our primary contributions. We strongly hope that the reviewer would agree that the totality of these ideas and results make our work a significant and novel contribution to the field.
>
> (2) We believe our core message was obscured by technical details and ad-hoc experimental choices, however this was a legacy of many experimental iterations with large models and limited compute/time. Since our original submission we have continued work and shown that much of the obfuscating complexity is not necessary for Poppy to work. Concretely:
>
> * Poppy can be trained in a simple two-stage process. First a single agent is trained with a standard objective to efficiently learn a shared encoder. Secondly, the decoder is cloned K times to initialize the population of agents which is then trained with the Poppy objective. No further complexity is required (and this new process considerably reduces the population training time by ~50% with no loss of final performance).
> * Whilst our original work applied Poppy differently to TSP compared to Knapsack and CVRP, this domain-specific modification was orthogonal to our core contribution and we have since found it unnecessary. Concretely, we now run the algorithm the same way for all problem types and still obtain SOTA results.
>
> We have updated our manuscript to reflect these simplifications (see Sec 3.2 in particular), which we believe to be a substantial improvement.
>
> > “How is the framework similar/different from e.g. describing evaluation objective as a maximum of sampled trajectories from a (potentially very expressive) policy? In the $K=2$  case, $\pi_{\theta_{meta}}$ for example, could be a very bimodal policy that can also produce diverse
> behaviors from its two modes.”
>
> In principle one could consider a single policy that is highly expressive with multiple behavioral modes, however in practice it is not clear that a typical RL training process is well suited to produce such a policy. Whilst a single agent could be trained on only the best trajectory from several samples on each problem instance, whether this would easily lead to specialized behavioral modes adapted to different problems is unclear (e.g. given the shared parameters across the whole model). More practically, in our setting the bulk of the computation is done in the (deterministic) problem embedding by the large encoder, with per-step decoding (where action sampling would occur) consisting of a lightweight policy head. As such, it is unlikely that the decoder could be sufficiently expressive to explore this idea without infeasible run-times. Ultimately, whilst this is an interesting question and direction of future work, it is separate from the core contributions of our work – (1) demonstrating that a specialized population of policies provides SOTA performance on CO problems and (2) presenting a practical methodology for achieving this.

---

### Official Review · Reviewer_rpzd · 2022-10-22

**Confidence:** 4
**Correctness:** 3
**Technical Novelty And Significance:** 3
**Empirical Novelty And Significance:** 2
**Recommendation:** 6

**Clarity, Quality, Novelty And Reproducibility:**

**Clarity & Quality**:
The paper is clear and very well-written, I enjoyed reading it. The idea of using a population of agents for CO is well motivated and illustrated on an intuitive example. Different plots (Fig 3) help to understand the effects of the proposed approach.

**Originality**:
Because of an existing closely related work [1], the technical originality is limited (see Weakness 1). In terms of experiments, the datasets and settings are standard.

**Reproducibility**:
The experimental setting is described in detail; the code and datasets are not provided but it looks like they would be after the double-blind reviewing period.

**Questions/remarks:**

1. In Algorithm 1, the two last lines, I could not see how these formulas allow to only update the parameters of the “best” decoder for each trajectory.

1. About the Training Procedure, I wonder if there is any intuition on how to fix the number of training steps of phase 2. If too small, the mentioned imbalance between the decoders is an issue; if too large I guess the decoders may become too similar. Aside from tuning this number experimentally, I wonder if there is any criteria that can be used to switch from phase 2 to 3.

1. A related work that's worth mentioning: [4].

[4] Kim et al, Learning Collaborative Policies to Solve NP-hard Routing Problems, Neurips 2021



**Strength And Weaknesses:**

**Strengths**
1. The paper is clear and very well-written.
1. The idea of using a population of agents is well motivated.
1. The proposed loss that enables an effective diversity without explicitly guiding the specialization seems general enough and could be used in other contexts.
1. Experimental evaluation shows strong results on the selected tasks.

**Weaknesses**
1. A very similar idea of training multiple decoders for the Attention Model was explored in MDAM [1] and applied to the TSP and CVRP:
   * As far as I can see, the main contribution/difference of the present paper w.r.t MDAM is the different loss -- which considerably limits the novelty of the paper.
   * Although the experimental results of the present paper seem stronger, this can also come from using POMO [2] training which is known to significantly improve the AM performance.
   * Including a precise discussion of the differences w.r.t to MDAM and including it in the experiments baselines would be useful.

1. The experimental evaluation focuses on datasets (synthetic TSP/CVRP instances with 100, 125 or 150 nodes and KP with 100 or 200 items) that do not seem challenging enough, today, for learning-based methods. Indeed:
    * The reported optimality gap of existing learning-based methods is already very small: less than 0.01% for TSP100 and 0.2% for TSP150, and less than 0.005% for the KP. These are already very good to excellent for heuristics in general.
    * In this context, even dividing the opt gap by 5 (as claimed in the abstract for TSP) does not seem very significant to me. This is especially visible in Table 3.
    * The scale of these problems is quite small compared to what other learning-based methods are able to solve (e.g. [3] solves TSP instances with up to 10,000 nodes, with supervision only on small graphs)
    * I would suggest to explore more challenging problems (e.g. larger sizes) to really showcase the experimental value of the proposed approach.

1. I can see the interest of having a population of agents in Fig 3;  however, (again) the gains seem very small:
    * in Fig 3 bottom left, we see that removing 1 agent from the population would lead to a deterioration of the opt gap of at most ~ 0.0004%.
    * This raises the question of is it because the opt gap is already very small (cf previous point) or is it the influence of the individual agents that is not significant (i.e. how different are the agents).
    * I wonder how, for example, taking the best of 4 samples x 4 agents would compare to greedily using 16 different agents.


[1] Xin et al, Multi-Decoder Attention Model with Embedding Glimpse for Solving Vehicle Routing Problems, AAAI 2021

[2] Kwon et al, POMO: Policy Optimization with Multiple Optima for Reinforcement Learning, Neurips 2020

[3] Fu et al, Generalize a Small Pre-trained Model to Arbitrarily Large TSP Instances, AAAI 2021



**Summary Of The Paper:**

The paper proposes Poppy, an RL-based approach to learn a population of constructive heuristics for combinatorial optimization problems.   The presented architecture is based on the encoder-decoder Attention Model, with a shared encoder and one decoder per agent of the population. Starting from a pretrained model, Poppy training is based on a policy gradient that only depends on the performance of the best agent for each instance; and serves to only update that agent’s parameters. Poppy is evaluated on three well-known CO problems: the TSP, CVRP and KP.

**Summary Of The Review:**

I would vote for reject, because of the combination of:

(i) The limited novelty, especially w.r.t [1]: since the population aspect has already been explored with a similar architecture for similar problems, the main novelty seems to be the loss -- although I encourage the authors to make a precise comparison and update their claims.

(ii) The experimental evaluation is not convincing because, in my opinion, it shows marginal gains on tasks where learning-based methods have already achieved excellent results. Evaluation on more challenging problems/settings (e.g larger instances) would greatly increase the empirical value of this paper.

---

> ### Author Response · Authors · 2022-11-14
> **Authors' response to Reviewer rpzd**
>
> We thank the reviewer for their detailed feedback and continued consideration of our work. Point-by-point responses to the points raised are provided below, and we are hopeful that these will address all concerns raised.
>
> > 1. “A very similar idea of training multiple decoders for the Attention Model was explored in MDAM [1]…”
>
> We were not aware of this paper, and thank the reviewer for this insight! Although the architecture (shared encoder with multiple decoders) is similar, we emphasize that in our case this is simply a means towards computational feasibility rather than our core contributions (Poppy would still be viable without shared encoders). We don’t believe this significantly harms novelty as our core focus is specialization of a population. In this vein, the training process is different: MDAM uses the KL divergence, which involves explicitly trading off performance with diversity. Furthermore, computing this KL divergence for the whole trajectory is intractable, and therefore is used only at the first timestep. In this sense, MDAM is very POMO-like (forcing diversity for the first action only). In contrast, our method drives diversity solely by maximizing population-level performance (with no explicit diversity metric/loss), and uses the whole trajectory. We also observe that using the KL divergence does not scale well with the population size, and that MDAM only evaluated populations up to 5 agents.
>
> We have included MDAM in the Related Work and highlighted these differences. We have also added MDAM as a baseline in TSP100 and CVRP100 to our tables. The authors did not originally evaluate MDAM in our other settings (TSP125, TSP150, CVRP125, CVRP150, KP100 and KP 200). Since the performance of MDAM is poorer than POMO’s and Poppy’s in TSP100 and CVRP100, we have decided to omit these experiments for the time being.
>
> > 2.a-b “The reported optimality gap of existing learning-based methods is already very small.”
>
> We believe that the performance improvement of Poppy is highly significant. We agree that for many CO problems, even simple policies can provide small performance optimality gaps, but in practice this means substantial research breakthroughs can look modest in absolute terms (especially with further improvements being increasingly hard to obtain as we near optimality). However:
> 1. *Poppy is as significant of an advance as prior SOTA breakthrough results.* Taking TSP100 as our example, Poppy (optimality gap 0.008%) closes 83.3% of the optimality gap compared to EAS (0.048%, Huttung et al, ICLR 2022). EAS closed 63% of the gap left by POMO (0.13%, Kwon et al. NeurIPS 2020), which in turn had closed 97% of the optimality gap w.r.t. AM (Kool et al., ICLR 2019).
>
> 2. *Poppy provides significant efficiency improvements.*  For practical applications, performance is not just measured by the obtained objective value, but also by the time taken to acquire it. In this vein, Poppy is much faster than EAS, the current SOTA (e.g. 9 min. instead of 5 hours on TSP100) as the latter relies on active search (fine-tuning) at test time.
> 3. *The lower optimality gap Poppy translates into regular per-instance improvements.* An alternative way to compare two approaches (that may be more aligned with practical application) is in how often they outperform each other on random instances. Compared to the SOTA fast RL method POMO (i.e. solving instances given roughly the same wall-clock time) Poppy 16 is strictly better in 81.67% and 34.30% of the instances for TSP100 and KP100, and better or equal in 98.02% and 99.95%, whereas Poppy 32 is strictly better in 93.83% of the instances for CVRP100 and better or equal in 93.87% (see Sec. 4.1-4.3 in paper).
>
> > 2.c “The scale of these problems is quite small compared to what other learning-based methods are able to solve (e.g. [3])...”
>
> We agree that a parallel line of work is focussed on scaling to larger instances, however the RL methods we consider (Poppy, POMO, EAS) are tailored for providing high performance on smaller solutions as they construct solutions sequentially. Whilst our TSP instances can be tackled by established solvers, e.g. LKH3, Poppy still provides near optimal performance in less time (see Table 1). We also emphasize that this is not the case for CVRP for which 100 customers is already considered challenging (as demonstrated by the results of LKH3 in Table 2, which we outperform without any finetuning on test instances).
>
> With regards to Fu et al, (ref [3] from the reviewer) this is a TSP specific approach designed for scalability. We find that on TSP100 Poppy is notably stronger (0.008% vs 0.037%); highlighting the different performance vs scaling paradigms. We agree this is relevant context and have extended our discussion of Fu et al in the related work, added the TSP100 comparison to Table 1 and commented on the scalability of methods in our empirical comparison (see Results in Sec 4.1).

---

> > ### Author Response · Authors · 2022-11-16
> > **Authors' response to Reviewer rpzd [Part 2/2]**
> >
> > > 3. “I can see the interest of having a population of agents in Fig 3; however, (again) the gains seem very small… In Fig 3 bottom left, we see that removing 1 agent from the population would lead to a deterioration of the opt gap of at most ~0.0004% [what is the reason behind this?]”
> >
> > A deterioration of 0.0004% represents a deterioration of 5% in comparison to the gap achieved by Poppy 16. So we would like to emphasize that this small deterioration is not linked to agents being redundant, but to the gap being already small. The relevance of each member of the population is further highlighted by the improving performance of Poppy 4, 8 and 16. To make this point clearer, we added a new baseline in the experiments: an ensemble of 16 agents trained with POMO (thus not specialized). We show that Poppy consistently outperforms this baseline across all our environments. We also performed the experiment of comparing taking 4 random samples of Poppy 4 and Poppy 16 on the TSP:
> >
> > | Method                           | Tour length | Opt. gap | Time |
> > |----------------------------------|------------------------|------- |------ |
> > | Optimal			     | 7.7645      |    -     | -      |
> > | Poppy 4 (4 samples) 		     | 7.7658      | 0.016 %  | 9M 	  |
> > | Poppy 4 (greedy)    		     | 7.767       | 0.029% % | 2M     |
> > | Poppy 16 (greedy)   		     | 7.7651      | 0.008%   | 9M 	  |
> >
> >
> > Poppy 4 with 4 random samples is better than Poppy 4 (greedy), but outperformed by Poppy 16 by a large margin for the same inference time. It indeed shows that going for a larger population instead of more stochastic samples is beneficial.
> >
> > > Q1. In Algorithm 1, the two last lines, I could not see how these formulas allow to only update the parameters of the “best” decoder for each trajectory.
> >
> > The specialization occurs in line 7: only the trajectories from the set of best trajectories $\mathcal{T}^*$are used to backpropagate the loss. We have modified Algorithm 1 to make this selection step clearer.
> >
> > > Q2. “About the Training Procedure, I wonder if there is any intuition on how to fix the number of training steps of phase 2…”
> >
> > This is a very interesting question, as we noticed that Poppy was robust to the number of training steps used in phase 2, to the point where we realized phase 2 could be removed altogether: this was a legacy of many experimental iterations with large models and limited compute/time. Concretely, Poppy can be trained in a simple two-stage process. First a single agent is trained with a standard objective to efficiently learn a shared encoder. Secondly, the decoder is cloned K times to initialize the population of agents which is then trained with the Poppy objective. No further complexity is required (and this new process considerably reduces the population training time by ~50% with no loss of final performance).
> >
> > We have updated our manuscript to reflect this simplification (see Sec 3.2 in particular), which we believe to be a substantial improvement.
> >
> > > Q3. A related work that's worth mentioning: [4].
> > [4] Kim et al, Learning Collaborative Policies to Solve NP-hard Routing Problems, Neurips 2021
> >
> > This paper proposes a hierarchical strategy where a seeder proposes solution candidates, and a reviser refines parts of the proposed candidates. We agree that this reference is worth mentioning, and have thus added it to the related work.

---

### Official Review · Reviewer_5SRy · 2022-10-25

**Confidence:** 4
**Correctness:** 3
**Technical Novelty And Significance:** 3
**Empirical Novelty And Significance:** 3
**Recommendation:** 5

**Clarity, Quality, Novelty And Reproducibility:**

My evaluation has 4 levels: Excellent, good, fair, poor.

Clarity: fair. The claim that the method is "theoretically grounded" seems not to be supported. The lower bound of the objective is too weak to be a theory and people generally think of a gaurantee of some kind of performance when you say that.

Quality: fair.

Novelty: Excellent. This is a novel idea!

Reproducibility: Fair. The description of the method seems clear and the method itself (the key idea) is not complicated. But there are some engineering tricks that make it a bit challenging to reproduce.

**Strength And Weaknesses:**

Strengths:
1) Using RL for CO problems is an interesting topic and of practical importance.
2) The idea of training a (diverse) population agents for CO problems is interesting and novel.
3) The rationale behind POPPY is well-motivated with the discusison of the line of related works, as well as using the simple example in Figure 1.
4) It shows considerable improvement over SOTA RL methods on the TSP benchmark task.

Weaknesses:
1) The empirical evaluation is not entirely convincing --
1a) The improvements over SOTA RL methods like POMO are generally small on all of the three tasks. POMO itself is already pretty close to optimal solution, so that the edge of POPPY is barely significant (it is unclear if the authors have done t-tests). The evaluation would be more convincing if the authors can show that POPPY has more significant improvements.
1b) On the CVRP task, POPPY is outperformed by EAS.
1c) The baselines are missing for the KP task, making it even more challenging to evaluate the performance of POPPY.

2) There are many engineering heuristics to make the system work (e.g., when to freeze or unfreeze the encoder, whether or not the optimal solution depends on the starting point). Therefore, the overall performance may be fragile to the specific training architecture. Especially that the edge of POPPY seems to be small compared to the baselines, so that it is questionable whether these kind of improvements are due to engineering tricks or structured improvements due to maintaining a population of agents. Also, the results may be hard to reproduce.

3) It is unclear to me where the "diversity" in the population of agents lies in the objective function. It appears that POPPY is training a large set of agents with different policies, so that one of them "happens" to be good at a certain problem instance. As a fair comparison, perhaps you should separately train 16 POMO/LIH/EAS, and compare POPPY-16 with the best performance of those 16 policies. It is questionable whether POPPY can beat such a strong baseline.

**Summary Of The Paper:**

The paper is in the line of works that uses reinforcement learning for combinatorial optimization (CO) problems. The main contribution is the idea of training a population of complementary agents for a given distribution of CO problems, with the objective function being optimizing the performance of the best agent instead of the average. The proposed training approach, POPPY, is evaluated on three benchmark tasks, including TSP, CVRP, and KP. It shows state-of-the-art performance on the TSP task.

**Summary Of The Review:**

The idea of training a diverse population of agents to solve CO problems is generally interesting, novel, and well-motivated. But it is not clear where diversity is encouraged, and the evaluation is not entirely convincing.

---

> ### Author Response · Authors · 2022-11-14
> **Authors' response to Reviewer 5SRy [Part 2/2]**
>
> > 2. “There are many engineering heuristics to make the system work…”
>
> We agree that our core message was obscured by technical details and ad-hoc experimental choices.  Since our original submission we have continued work and observed that the heuristics listed by the reviewer are not necessary for Poppy to work. Specifically;
>
> 1. Poppy can be trained in a simple two-stage process (i.e. without (un)freezing the encoder). First a single agent is trained with a standard objective to efficiently learn a shared encoder. Secondly, the decoder is cloned K times to initialize the population of agents which are trained with the Poppy objective. This process reduces the population training time (by ~50%) with no loss, or even slight improvement, in performance.
>
> 2. We now treat starting points the same way for all problem types and still obtain SOTA results (i.e. train only the best agent on each [problem, starting point] pair). Previously, for TSP we trained only the best agent on the best starting point for each problem (as the optimal solution is independent of the starting point), however this domain-specific tweak is orthogonal to our contribution and has since been shown to not provide a performance boost for larger populations.
>
> We have updated the paper to reflect these simplifications (see Sec 3.2) and moved details not relevant to our core contributions to the appendix. This has the significant benefit of presenting a straightforward and consistent application of Poppy’s core idea: learning a population of specialized policies. We believe this is a substantial improvement in our manuscript and are grateful to the reviewer for highlighting this point.
>
> > 3. “It is unclear to me where the "diversity" in the population of agents lies in the objective function…”
>
> It is not the case that the Poppy objective results in an ensemble of agents that only have differing performance on particular problems by chance. By training only the best agent on each problem instance, an individual agent sees only a subset of the overall training distribution. Specialization emerges because these subsets are by definition problem instances on which a specific policy can reliably outperform others. The only way in which these “specialized” agents would not be diverse is if the best policy on all subsets of the distribution is the same (i.e. a single policy is optimal for the entire distribution). We motivate why this is not the case for complex CO problems in Sec 3.1 and comment directly on the above in Sec 3.2.
>
> We agree with the reviewer that POMO+ensemble is a natural baseline (it corresponds to keeping the same model architecture but removing the specialization objective). We have added these results in the paper (Tables 1, 2 and 3) and confirm that Poppy significantly outperforms this baseline on every task. This can be considered an ablation study in favor of our specialization objective. We note than LIH+ensemble and EAS+ensemble are not practical, as (1) both methods are “slow” RL approaches with hours of inference time for a single model and (2) in principle Poppy is an orthogonal contribution to these methods (i.e. we could run EAS or LIH on top of a specialized population, albeit with considerable computational overhead).
>
> > “The claim that the method is "theoretically grounded" seems not to be supported…”
>
> By showing that our training objective lower bounds the expected population-level performance, we aimed to provide a theoretical justification of our intuition for the Poppy objective, however we agree that this is not equivalent to strict performance guarantees. To prevent misunderstandings we have made the following changes.
> 1. Removed “theoretically grounded” from the abstract.
> 2. Provided extended analysis of the bias in our stochastic approximation of the gradient of the Poppy objective (see App. B.1-B.2). In brief, this bias vanishes in the limit of deterministic agents or perfect specialization, which in practice means the gradient is slightly biased, but that this decreases as training progresses.
> 3. Emphasized throughout our paper that Poppy does not provide any diversity or performance guarantee (e.g. “Population-Based Training Objective” in Sec. 3.2). However, we also note that, as discussed above, diversity does maximize our objective in the highly probable case that a single agent is not optimal on all subsets of the training distribution.
>
> We believe these modifications strike the right balance between investigating the theoretical implications of Poppy whilst not detracting from the primary emphasis on the SOTA empirical performance of specialized populations of agents.
>
> > “…There are some engineering tricks that make it a bit challenging to reproduce”.
>
> We have provided a link to source code, https://anonymous.4open.science/r/poppy-6D20 in the revision. With regards to “engineering tricks”, we refer to our answer to your second point as these have largely been removed or simplified.

---

> > ### Comment · Reviewer_5SRy · 2022-12-09
> > **Not changing my mind**
> >
> > Thank the authors for the responses.
> > Based on the authors' responses and the comments of the other reviewers, I would like to maintain my evaluation.

---

> ### Author Response · Authors · 2022-11-14
> **Authors' response to Reviewer 5SRy [Part 1/2]**
>
> We thank the reviewer for their detailed feedback. We are heartened by the positive assessment of our novelty and believe that the reviewers' concerns can, and have, be directly addressed. Point-by-point responses are provided below.
>
> > 1.a “The improvements over SOTA RL methods like POMO are generally small on all of the three tasks…”
>
> We believe that in both the context of the field and in absolute statistical terms, the performance improvement of Poppy is highly significant. First, we note that for combinatorial problems, even simple policies can provide strong performance (e.g. the optimality gap of a greedy strategy on Knapsack is 0.125% as per Table 3), so even substantial breakthroughs can look modest in absolute terms, especially with further improvement being increasingly hard to obtain as we near optimality. Concretely, the table below compares Poppy to the prior 3 SOTA methods (EAS, POMO, AM) on TSP100 and we see that the performance improvement of Poppy w.r.t. EAS is as large as that of POMO vs AM, and EAS vs POMO. Moreover, Poppy beats EAS, the current SOTA, in a fraction of the time (9 min. instead of 5 hours) as the latter relies on active search (fine-tuning) at test time.
>
> In terms of statistical significance, we note that our test sets are very large (1000 to 10000 instances) with the same set of randomly generated instances used to test every method (i.e. there is no random variance related to instance generation). In line with all prior work, confidence intervals are not reported and we believe a better metric is the rate at which Poppy outperforms POMO (i.e. the current SOTA “fast” RL method). Specifically, Poppy 16 is strictly better in 81.67% and 34.3% of the instances for TSP100 and KP100, and better or equal in 98.02% and 99.95%, whereas Poppy 32 is strictly better in 93.83% of the instances for CVRP100. These comparisons have been added to the results sections of our revised manuscript.
>
> | Method                           | Tour length (opt. gap) | Percentage of opt. gap closed |
> |----------------------------------|------------------------|-------------------------------|
> | AM (Kool et al., ICLR 2019)      | 8.12 (4.53%)       	| -                             |
> | POMO (Kwon et al., NeurIPS 2020) | 7.774 (0.13%)      	| 97.13%                        |
> | EAS (Hottung et al., ICLR 2022)  | 7.768 (0.048%)     	| 63.08%                        |
> | Poppy 16                         | 7.765 (0.008%)     	| 83.33%                        |
>
> > 1.b “On the CVRP task, POPPY is outperformed by EAS”
>
> We emphasize that Poppy and EAS use orthogonal approaches and that comparisons should not only consider performance, but also the time budget and practical utility of these methods. Concretely, Poppy is a sampling-only method able to produce high quality results quickly, whereas EAS finetunes the policy at inference requiring considerably longer running time. For example, on TSP100 Poppy outperforms EAS in only 9 mins, compared to 5 hours for EAS (even longer than the 82 mins required for an exact TSP solver). Whilst on CVRP in particular, finetuning seems to provide significant gains at the cost of longer inference times, Poppy is still the clear SOTA for “fast” RL methods (see Tables 1-3). In any case, we note that EAS’ active search and Poppy’s population-based training are not mutually exclusive, both breakthroughs are independently valuable as it appears possible to combine them for further improvements.
>
> > 1.c “The baselines are missing for the KP task”
>
> We added a new baseline for KP: an ensemble of 16 agents trained with POMO [1] instead of Poppy, which we outperform consistently. With the exception of attention-based RL models, none of the baselines we use for TSP (Table 1) or CVRP (Table 2) are applicable to KP (much of the prior literature has focused on routing challenges). We benchmark against POMO as it is the SOTA RL algorithm for KP (EAS does not tackle this problem in its paper or open-source implementation). Indeed, from Table 4 in the POMO paper we see that the only other baselines are weaker RL predecessors [2, 3] - which we do not feel are useful baselines. Whilst we would be more than happy to include baselines that the reviewer may specifically feel are lacking, we note that our results already show a clear improvement of Poppy over the prior SOTA and that the inclusion of KP is primarily to highlight that our method is not restricted to routing problems.
>
> [1] Kwon et al. POMO: Policy Optimization with Multiple Optima for Reinforcement Learning. NeurIPS 2020.
> [2] Kool et al. Attention, Learn to Solve Routing Problems!. ICLR 2019.
> [3] Vinyals et al. Pointer networks. NeurIPS 2015.

---

### Author Response · Authors · 2022-11-23
**Author's general response**

We thank all the reviewers for their insightful comments that helped to improve the general quality of this work. We made great efforts to update the manuscript accordingly. In particular:
- Poppy can be now trained in a simple two-stage process (Sec 3.2).
- Whilst our original work applied Poppy differently to TSP, we have since found it unnecessary. Concretely, we now run the algorithm the same way for all problem types and still obtain SOTA results (Sec 3.2).
- We provided an extended analysis of the bias in our stochastic approximation of the gradient of the Poppy objective. In brief, this bias vanishes in the limit of deterministic agents or perfect specialization, which in practice means the gradient is slightly biased, but that this decreases as training progresses (App. B.1-B.2).

We hope that other specific concerns and remarks were fully answered in our responses, as well as the changes made to the manuscript, and that these are sufficient to reconsider their assessments. Thanks to the OpenReview format, we would be delighted to initiate a discussion with the reviewers if some elements were not addressed satisfactorily.

---

> ### Author Response · Authors · 2022-11-30
> **Follow Up Discussion**
>
> Dear Reviewers,
>
> We hope our answers and the revised version of the paper have addressed your concerns. We strongly believe that your comments have improved the quality of the paper.  We appreciate your reconsideration and, in the case that you are still do not feel able to fully endorse our work, would be grateful to start a discussion of any unresolved aspects.
>
> Best,
> Authors

---

### Decision · Program_Chairs · 2023-01-20

**Decision:**

Reject

**Justification For Why Not Higher Score:**

NA

**Justification For Why Not Lower Score:**

NA

**Metareview: Summary, Strengths And Weaknesses:**

This paper presents a new ensemble method for solving combinatorial optimization problems using RL. While the studied problem is a relevant and interesting application of RL, all reviewers agree that the technical novelty of the proposed algorithm is limited, and the experimental evaluation is weak: (1) the evaluated tasks are too simple, they are small-size synthetic problems; (2) the proposed method only slightly outperforms baselines, even many hand-tuning. I vote for rejection.